# Neurons enhance blood–brain barrier function via upregulating claudin-5 and VE-cadherin expression due to glial cell line-derived neurotrophic factor secretion

Lu Yang, Zijin Lin, Ruijing Mu, Wenhan Wu, Hao Zhi, Xiaodong Liu*, Hanyu Yang*, Li Liu*

Department of Pharmacology, School of Pharmacy, China Pharmaceutical University, Nanjing, China

**\*For correspondence:**
xdliu@cpu.edu.cn (XL);
shenyhy@cpu.edu.cn (HY);
liulee@cpu.edu.cn (LL)

**Competing interest:** The authors declare that no competing interests exist.

## eLife assessment

This revised study presents **valuable** evidence that a combination of endothelial cells, astrocytes, and neuroblastoma cells of human origin can integrate to form an *in vitro* brain blood barrier, that recapitulates key aspects of its natural counterpart, especially at short times. **Convincingly**, the mechanism by which neuroblastoma-secreted GDNF increases Claudin-5 and VE-cadherin is described. To substantiate the role of GDNF in vivo, authors demonstrated that knock-down of this neurotrophic factor, increased the permeability of the brain blood barrier in mice. This *in vitro* system can be used to study the permeability of the human brain blood barrier to different drugs.

**Abstract** Blood–brain barrier (BBB) prevents neurotoxins from entering central nervous system. We aimed to establish and characterize an *in vitro* triple co-culture BBB model consisting of brain endothelial cells hCMEC/D3, astrocytoma U251 cells, and neuroblastoma SH-SY5Y cells. Co-culture of SH-SY5Y and U251 cells markedly enhanced claudin-5 and VE-cadherin expression in hCMEC/D3 cells, accompanied by increased transendothelial electrical resistance and decreased permeability. Conditioned medium (CM) from SH-SY5Y cells (S-CM), U251 cells (U-CM), and co-culture of SH-SY5Y and U251 cells (US-CM) also promoted claudin-5 and VE-cadherin expression. Glial cell line-derived neurotrophic factor (GDNF) levels in S-CM and US-CM were significantly higher than CMs from hCMEC/D3 and U-CM. Both GDNF and US-CM upregulated claudin-5 and VE-cadherin expression, which were attenuated by anti-GDNF antibody and GDNF signaling inhibitors. GDNF increased claudin-5 expression via the PI3K/AKT/FOXO1 and MAPK/ERK pathways. Meanwhile, GDNF promoted VE-cadherin expression by activating PI3K/AKT/ETS1 and MAPK/ERK/ETS1 signaling. The roles of GDNF in BBB integrity were validated using brain-specific *Gdnf* silencing mice. The developed triple co-culture BBB model was successfully applied to predict BBB permeability. In conclusion, neurons enhance BBB integrity by upregulating claudin-5 and VE-cadherin expression through GDNF secretion and established triple co-culture BBB model may be used to predict drugs' BBB permeability.

## Introduction

As a dynamic interface between the blood circulatory system and the central nervous system (CNS), the blood–brain barrier (BBB) maintains homeostasis and normal function of CNS by strictly regulating material exchange between the blood and brain (*Palmiotti et al., 2014*). The maintenance of BBB is mainly attributed to the expression of tight junctions (TJ) and adherent junctions (AJ) between

adjacent brain endothelial cells and a variety of drug transporters. However, as a double-edged sword that protects CNS function, BBB also restricts the transport of some drugs from blood to brain, leading to poor CNS therapeutic effects and even CNS treatment failure (*Banks, 2016*).

Several *in silico*, *in vitro*, *in situ*, and *in vivo* methods have been developed to assess the permeability of drugs across BBB, but each method has its limitations (*Hanafy et al., 2021*). The *in situ* brain perfusion (ISBP) is considered the 'gold standard' for assessing BBB permeability but there exist limits related to animal ethics. Moreover, ISBP is not suitable for human or high-throughput workflows. Thus, a suitable, accurate, and high-throughput *in vitro* BBB model is required to predict the permeability of drug candidates through BBB.

BBB is formed by neurovascular units (NVU), composed of neural (neurons, microglia, and astrocytes) and vascular components (vascular endothelial cells, pericytes, and vascular smooth muscle cells) (*Potjewyd et al., 2018*). The constant crosstalk and interactions among these cells contribute to the structural and signaling-based regulation of transcellular and paracellular transport, control of BBB permeability, and regulation of cerebral circulation (*Arvanitis et al., 2020*; *Muoio et al., 2014*; *Potjewyd et al., 2018*). Although brain microvascular endothelial cells (BMECs) are commonly utilized as an *in vitro* BBB model to assess drug permeability across BBB, *in vitro* mono-culture of BMECs tends to lose some unique characteristics without the support of other cell types. To overcome the drawbacks of mono-culture BBB models, a range of multicellular co-culture BBB models co-cultured with other NVU elements (such as astrocytes or pericytes) have been developed. Astrocytes are the most abundant glial cell type in brain (*Clasadonte and Prevot, 2018*). Their terminal feet cover >80% of the surface of capillaries to form interdigitating coverage without slits (*Mathiisen et al., 2010*). In addition, astrocytes also contain several proteins related to the tight binding of the basement membrane. The *in vitro* multicellular co-culture BBB models composed of astrocytes, pericytes, and endothelial cells are considered to mimic the vascular structure and offer paracellular tightness in BBB (*Ito et al., 2019*; *Nakagawa et al., 2009*; *Watanabe et al., 2021*).

Neurons also play a crucial role in the NVU and may be involved in the regulation of BBB function. The maturation of BBB in mice considerably overlaps with the establishment of neuronal activity (*Biswas et al., 2020*). A study revealed that the conditioned medium (CM) collected from primary rat neurons also attenuated cell death caused by glucose–oxygen–serum deprivation (*Lin et al., 2016*). Furthermore, the primary rat neurons were reported to affect the differentiation and formation of BMECs (*Savettieri et al., 2000*; *Schiera et al., 2003*; *Xue et al., 2013*). Recent research reported that, compared with the double co-culture of hCMEC/D3 and 1321N1 (human astrocytoma cells), the triple co-culture of hCMEC/D3, 1321N1, and human neuroblastoma SH-SY5Y cells exhibited a higher transendothelial electrical resistance (TEER) (*Barberio et al., 2022*). A similar report showed that co-cultivation of RBE4.B (rat brain capillary endothelial cells) and neurons resulted in the lower permeability of [$^3$H] sucrose than RBE4.B cells grown alone (*Schiera et al., 2005*). All these studies indicate that neurons, as the elements of the NVU, may be tightly connected to the formation and maintenance of BBB functions.

The aims of the study were: (1) to establish and characterize an *in vitro* triple co-culture BBB model consisting of human brain endothelial cells (hCMEC/D3), human astrocytoma cells (U251), and human neuroblastoma cells (SH-SY5Y); (2) to investigate whether neurons were involved in the formation and maintenance of BBB integrity and explore their underlying mechanisms. The integrity of BBB was evaluated by quantifying the leakage of both fluorescein and FITC-Dextran 3–5 kDa (FITC-Dex); (3) to validate the *in vitro* results through *in vivo* experiments. Finally, the *in vivo*/*in vitro* correlation assay was analyzed to prove that the triple co-culture BBB model could better predict the BBB penetration of CNS drugs compared with the mono-culture BBB model.

## Results

### Establishment and characterization of the *in vitro* triple co-culture BBB model

Four types of BBB *in vitro* models were established to compare the contributions of U251 and SH-SY5Y cells to hCMEC/D3 cells (*Figure 1A*). TEER values were measured during the co-culture (*Figure 1B*). TEER values of the four *in vitro* BBB models gradually increased until day 6. On day 7, the TEER values showed a decreasing trend. Thus, 6-day co-culture period was used for subsequent

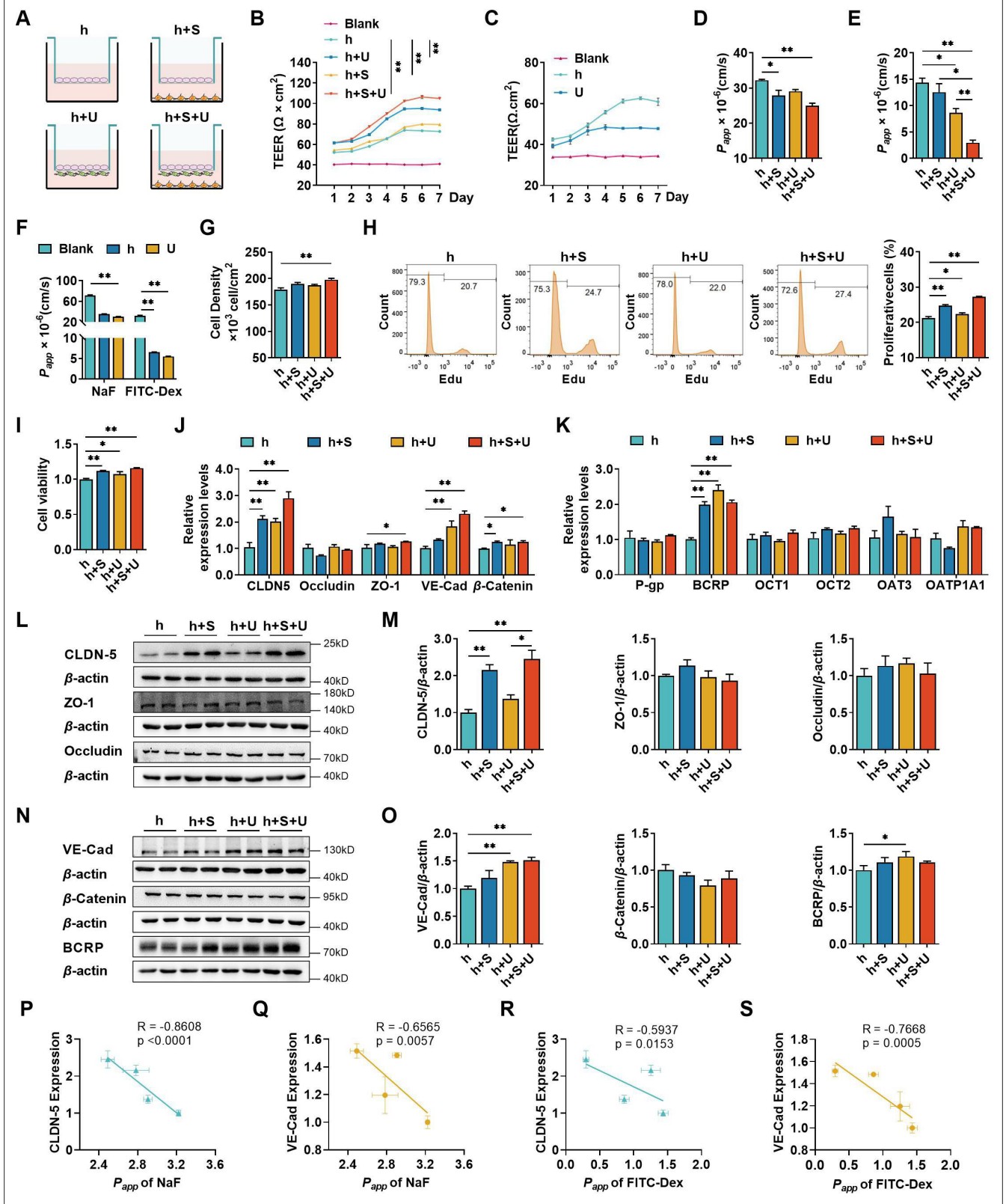

**Figure 1.** The effects of co-culture with U251 and/or SH-SY5Y cells on the integrity of hCMEC/D3 and blood–brain barrier (BBB) function. (**A**) Four different types of BBB models were prepared from hCMEC/D3 cells (h), SH-SY5Y cells (S), and U251 cells (U). (**B**) The transendothelial electrical resistance (TEER) of four models, and the TEER values in day 6 were compared. Blank: no cells. Four biological replicates per group. (**C**) The TEER of hCMEC/D3 and U251 cells monolayer. Four biological replicates per group. (**D, E**) The apparent permeability coefficient ($P_{app}$, ×10⁻⁶ cm/s) of fluorescein

*Figure 1 continued*

(NaF) and FITC-Dextran 3–5 kDa (FITC-Dex) of four BBB models. Four biological replicates per group. (**F**) The $P_{app}$ (×10$^{-6}$ cm/s) of NaF and FITC-Dex across the blank inserts, and hCMEC/D3 or U251 mono-culture models. Four biological replicates per group. The cell density (**G**), EdU incorporation (**H**) of hCMEC/D3 cells after mono/co-culturing. Three biological replicates per group. (**I**) Cell viability of hCMEC/D3 cells after mono/co-culturing. Four biological replicates per group. (**J, K**) The mRNA levels of tight junction proteins, adherent junction proteins, and transporters. Four biological replicates per group. The protein expression levels of claudin-5 (CLDN-5), ZO-1, occluding (**L, M**), VE-cadherin (VE-Cad), *β*-catenin, and BCRP (**N, O**) in hCMEC/D3 cells. Four biological replicates per group. The correlations between the $P_{app}$ (×10$^{-5}$ cm/s) of NaF and claudin-5 expression (**P**), or VE-cadherin expression (**Q**). The correlation between $P_{app}$ (×10$^{-5}$ cm/s) of FITC-Dex and claudin-5 expression (**R**), or VE-cadherin expression (**S**). The above data are shown as the mean ± SEM. For J and K, two technical replicates per biological replicate. One technical replicate per biological replicate for the rest. *p < 0.05; **p < 0.01 by one-way ANOVA test followed by Fisher's LSD test, Welch's ANOVA test, or Kruskal–Wallis test. The simple linear regression analysis was used to examine the presence of a linear relationship between two variables.

The online version of this article includes the following source data and figure supplement(s) for figure 1:

**Source data 1.** The western blot raw images in *Figure 1*.

**Source data 2.** The labeled western blot images in *Figure 1*.

**Source data 3.** Excel file containing summary data and data analysis of *Figure 1*.

**Figure supplement 1.** The induced proliferation of hCMEC/D3 cells by basic fibroblast growth factor (bFGF) slightly reduced the permeability of cell layers.

experiments. The highest TEER values were observed in the triple co-culture BBB model, followed by double co-culture with U251 cells and double co-culture with SH-SY5Y cells BBB models, hCMEC/D3 cells mono-culture BBB model showed the lowest TEER values. We also measured the TEER values of U251 monolayer cells, and the results showed that U251 cells themselves also contributed to the physical barrier of the model (*Figure 1C*). The apparent permeability coefficient ($P_{app}$) values of fluorescein and FITC-Dex were measured to characterize the integrity of four BBB models (*Figure 1D, E*). Consistent with the TEER values, the triple co-culture BBB model showed the lowest permeability of fluorescein and FITC-Dex, followed by double co-culture with U251 cells and double co-culture with SH-SY5Y cells BBB models. It was noticed that monolayer of U251 cells itself also worked as a barrier, preventing the leakage of permeability markers, which may explain why the permeability of FITC-Dex in double co-culture model with U251 cells is lower than that in double co-culture model with SH-SY5Y cells (*Figure 1F*). Co-culture with SH-SY5Y, U251, and U251 + SH-SY5Y cells also enhanced the proliferation of hCMEC/D3 cells. Moreover, the promoting effect of SH-SY5Y cells was stronger than that of U251 cells (*Figure 1G–I*). Furthermore, hCMEC/D3 cells were incubated with basic fibroblast growth factor (bFGF), which promotes cell proliferation without affecting both claudin-5 and VE-cadherin expression (*Figure 2F*). The results showed that incubation with bFGF increased cell proliferation and reduced permeabilities of fluorescein and FITC-Dex across hCMEC/D3 cell monolayer. However, the permeability reduction was less than that by double co-culture with U251 cells or triple co-culture. These results inferred that contribution of cell proliferation to the barrier function of hCMEC/D3 was minor (*Figure 1—figure supplement 1*).

The paracellular barrier of BBB is also associated with TJs, AJs, and transporters (*Abbott, 2013*). The mRNA levels of TJs (claudin-5, ZO-1, and occludin), AJs (VE-cadherin and *β*-catenin), and transporters (P-gp, BCRP, OCT-1, OCT-2, OAT-3, and OATP1A1) in hCMEC/D3 cells from the four BBB models were analyzed using quantitative real-time PCR (qPCR) (*Figure 1J, K*). Compared with hCMEC/D3 cell mono-culture model, double co-culture with SH-SY5Y, double co-culture with U251, and triple co-culture BBB models showed markedly increases in claudin-5, VE-cadherin, *β*-catenin, and BCRP mRNA expression. Expression of corresponding proteins was measured using western blot (*Figure 1L–O*). Notably increased claudin-5 expression was detected in double co-culture with SH-SY5Y cells and triple co-culture BBB models, while VE-cadherin expression was markedly increased in double co-culture with U251 cells and triple co-culture BBB models. Expression levels of other TJ proteins (ZO-1 and occludin) and AJ protein (*β*-catenin) were unaltered. The expression of BCRP was slightly affected by co-cultivation with U251 cells. Significant negative correlations were found between $P_{app}$ values of fluorescein and the expression of claudin-5 or VE-cadherin. $P_{app}$ values of FITC-Dex were also negatively correlated to the expression levels of claudin-5 or VE-cadherin (*Figure 1P–S*). These results indicate that the decreased permeability of fluorescein and FITC-Dex mainly results from the upregulated expression of both claudin-5 and VE-cadherin.

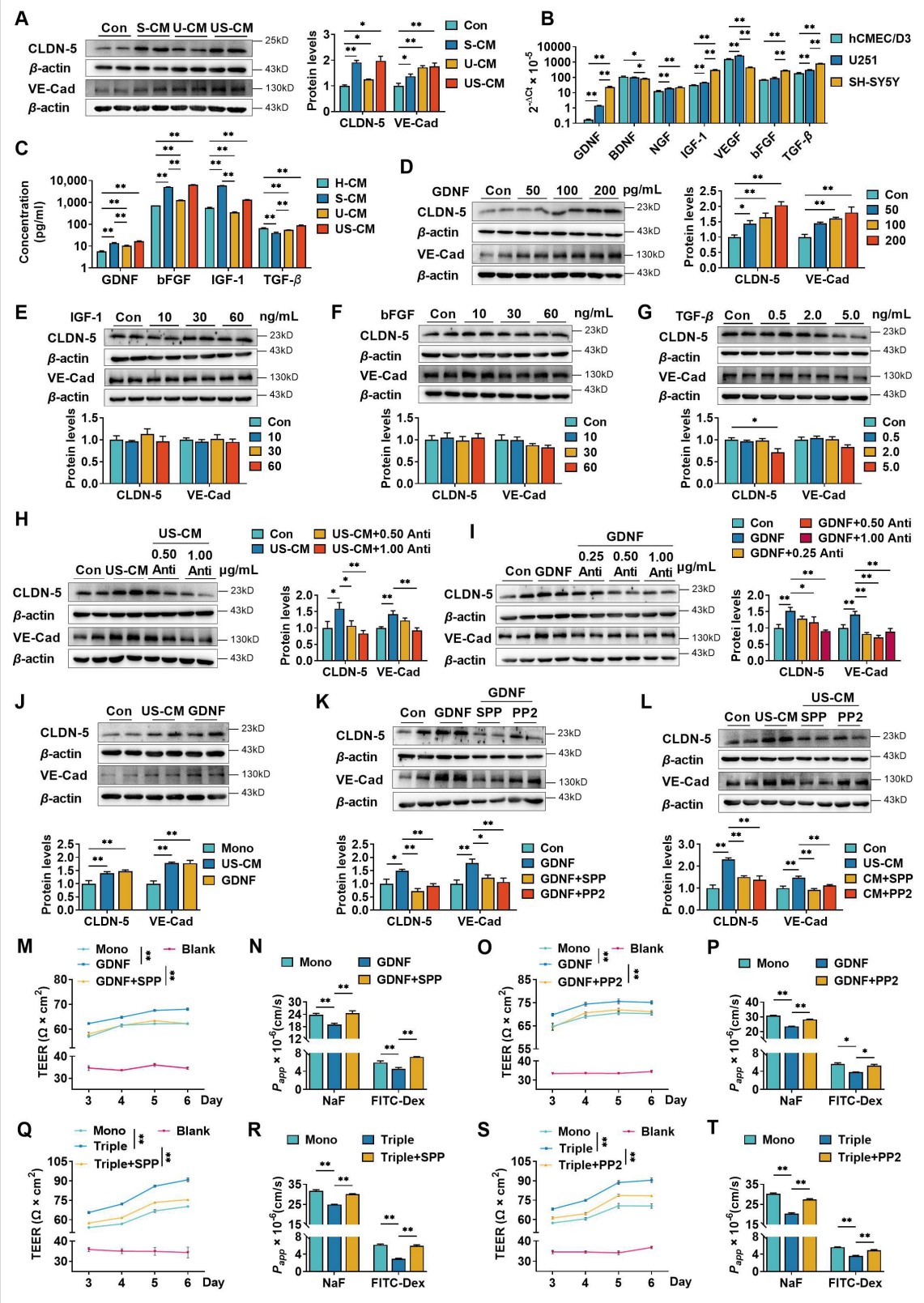

**Figure 2.** Neurons and astrocytes upregulated claudin-5 and VE-cadherin expression in hCMEC/D3 cells due to glial cell line-derived neurotrophic factor (GDNF) secretion. (**A**) Effects of conditioned medium (CM) on claudin-5 and VE-cadherin expression. Con: the normal medium; S-CM: the CM from SH-SY5Y cells; U-CM: the CM from U251 cells; US-CM: the CM from SH-SY5Y cells co-culture with U251 cells. (**B**) The mRNA expression levels of neurotrophic factors in hCMEC/D3, U251, and SH-SY5Y cells. (**C**) Concentrations of GDNF, basic fibroblast growth factor (bFGF), insulin-like growth

*Figure 2 continued on next page*

*Figure 2 continued*

factor-1 (IGF-1), and transforming growth factor-*β* (TGF-*β*) in the CMs. H-CM: the CM from hCMEC/D3 cells. Effects of GDNF (**D**), IGF-1 (**E**), bFGF (**F**), and TGF-*β* (**G**) on the expression of claudin-5 and VE-cadherin. The dosages have been marked in the figure. Effects of anti-GDNF antibody on the upregulation of claudin-5 and VE-cadherin expression induced by US-CM (**H**) or 200 pg/ml GDNF (**I**). (**J**) Effects of 200 pg/ml GDNF and US-CM on claudin-5 and VE-cadherin expression in primary rat brain microvascular endothelial cells. Effects of 3 µM RET tyrosine kinase inhibitor SSP-86 (SPP), and 5 µM Src family kinases inhibitor PP2 on the upregulation of claudin-5 and VE-cadherin induced by 200 pg/mL GDNF (**K**) and US-CM (**L**). Effects of SPP on the transendothelial electrical resistance (TEER) on day 6 (**M**), the permeability of NaF, and FITC-Dex (**N**) of the hCMEC/D3 mono-culture blood–brain barrier (BBB) model treating 200 pg/ml GDNF. Effects of PP2 on the TEER on day 6 (**O**), the permeability of NaF, and FITC-Dex (**P**) of the hCMEC/D3 mono-culture BBB model treating 200 pg/ml GDNF. Effects of SPP on the TEER on day 6 (**Q**), the permeability of NaF, and FITC-Dex (**R**) of the triple co-culture BBB model. Effects of PP2 on the TEER on day 6 (**S**), the permeability of NaF, and FITC-Dex (**T**) of the triple co-culture BBB model. The above data are shown as the mean ± SEM. Four biological replicates per group. For B and C, two technical replicates per biological replicate. One technical replicate per biological replicate for the rest. *$p < 0.05$; **$p < 0.01$ by one-way ANOVA test followed by Fisher's LSD test, Welch's ANOVA test, or Kruskal–Wallis test.

The online version of this article includes the following source data for figure 2:

**Source data 1.** The western blot raw images in *Figure 2*.

**Source data 2.** The labeled western blot images in *Figure 2*.

**Source data 3.** Excel file containing summary data and data analysis of *Figure 2*.

## Neurons and astrocytes upregulated the expression of claudin-5 and VE-cadherin by glial cell line-derived neurotrophic factor secretion

The hCMEC/D3 cells did not direct contact with U251 or SH-SY5Y cells in the double co-culture and triple co-culture BBB models, indicating that cell–cell interaction between U251, SH-SY5Y, and hCMEC/D3 cells relied on secreted active factors. To test this hypothesis, the effects of CM from SH-SY5Y cells (S-CM), U251 cells (U-CM), and co-culture of SH-SY5Y and U251 cells (US-CM) on the expression of claudin-5 and VE-cadherin in hCMEC/D3 cells were analyzed (*Figure 2A*). Both S-CM, U-CM, and US-CM markedly increased the expression of claudin-5 and VE-cadherin. US-CM showed the strongest induction effects on claudin-5 and VE-cadherin.

To investigate which cytokines were involved in the promotion of hCMEC/D3 cell integrity by U251 and SH-SY5Y cells, the mRNA expression levels of various cytokines in these three types of cells were compared. The results showed that U251 or SH-SY5Y cells exhibited significantly higher expression levels of glial cell line-derived neurotrophic factor (GDNF), nerve growth factor, insulin-like growth factor-1 (IGF-1), vascular endothelial growth factor, bFGF, and transforming growth factor-*β* (TGF-*β*) compared to hCMEC/D3 cells (*Figure 2B*). Furthermore, the mRNA expression of GDNF, IGF-1, TGF-*β*, and bFGF in SH-SY5Y cells was higher than those in U251 cells.

In these cytokines, GDNF (*Dong and Ubogu, 2018*; *Igarashi et al., 1999*; *Shimizu et al., 2012*), bFGF (*Shimizu et al., 2011*; *Wang et al., 2016*), IGF-1 (*Ko et al., 2009*; *Nowrangi et al., 2019*), and TGF-*β* (*Fu et al., 2021*) have been reported to promote BBB integrity. Thus, the concentrations of GDNF, bFGF, IGF-1, and TGF-*β* in the CMs were measured (*Figure 2C*). The results showed that levels of GDNF, bFGF, and IGF-1 in S-CM and US-CM were significantly higher than CMs from hCMEC/D3 cells (H-CM) and U-CM, but levels of TGF-*β* in S-CM and U-CM were lower than those in H-CM. Interestingly, the level of IGF-1 in US-CM was remarkably lower than that in S-CM, indicating that U251 cells suppressed IGF-1 secretion from SH-SY5Y. The effects of GDNF, bFGF, IGF-1, and TGF-*β* on the expression of claudin-5 and VE-cadherin were investigated (*Figure 2D–G*). Among the four tested neurotrophic factors, only GDNF induced the expression of claudin-5 and VE-cadherin in a concentration-dependent manner (*Figure 2D*). In contrast, a high level (5 ng/ml) of TGF-*β* slightly downregulated claudin-5 expression (*Figure 2G*). These results demonstrate that upregulation of claudin-5 and VE-cadherin expression by US-CM are attributed to secreted GDNF.

To provide additional verification of the deduction, anti-GDNF antibody was used to neutralize exogenous and endogenous GDNF in culture medium. Consistent with our expectation, the anti-GDNF antibody concentration-dependent reversed the US-CM-induced claudin-5 and VE-cadherin expression (*Figure 2H*). Furthermore, GDNF-induced upregulation of claudin-5 and VE-cadherin expression was also reversed by the anti-GDNF antibody (*Figure 2I*). The induction effects of US-CM and GDNF on the claudin-5 and VE-cadherin expression in hCMEC/D3 cells were also confirmed in primary rat BMECs (*Figure 2J*).

GDNF forms a heterohexameric complex with two GFRα1 molecules and two RET receptors to activate the GDNF–GFRα1–RET signaling (*Fielder et al., 2018*). The RET receptor tyrosine kinase inhibitor SPP-86 (*Bhallamudi et al., 2021*) and Src-type kinase inhibitor PP2 (*Morita et al., 2006*) were used to further investigate whether GDNF upregulated the expression of claudin-5 and VE-cadherin in hCMEC/D3 cells by activating the GDNF–GFRα1–RET signaling pathway. Both SPP-86 and PP2 markedly attenuated claudin-5 and VE-cadherin expression induced by GDNF (*Figure 2K*) and US-CM (*Figure 2L*).

The contributions of GDNF-induced claudin-5 and VE-cadherin expression to TEER and permeability were investigated using hCMEC/D3 cells mono-culture BBB model. GDNF significantly increased TEER values (*Figure 2M, O*) and decreased the permeability of fluorescein and FITC-Dex (*Figure 2N, P*), which were almost abolished by SPP-86 or PP2. Furthermore, treatment with SPP-86 or PP2 completely reversed the increased TEER values (*Figure 2Q, S*) and decreased permeability of fluorescein and FITC-Dex (*Figure 2R, T*) in the triple co-culture BBB model. These results indicate that neurons but also astrocytes upregulate claudin-5 and VE-cadherin expression in hCMEC/D3 cells by secreting GDNF. Subsequently, GDNF induces claudin-5 and VE-cadherin expression by activating GDNF–GFRα1–RET signaling.

## GDNF-induced claudin-5 and VE-cadherin expression of hCMEC/D3 by activating the PI3K/AKT and MAPK/ERK pathways

GDNF exerts its biological activities by activating several signaling pathways, including the phosphatidylinositol-3-kinase (PI3K)/protein kinase B (AKT), mitogen-activated protein kinase (MAPK)/extracellular regulated kinase (ERK), MAPK/c-Jun N-terminal kinase (JNK), and MAPK/ p38 pathways (*Fielder et al., 2018*). The effects of the PI3K/AKT, MAPK/ERK, MAPK/JNK, and MAPK/p38 pathway inhibitors LY294002 (*Figure 3A*), U0126 (*Figure 3B*), SP600125 (*Figure 3C*), and SB203580 (*Figure 3D*), respectively, on GDNF-induced claudin-5 and VE-cadherin expression in hCMEC/D3 cells were investigated. GDNF increased claudin-5 and VE-cadherin expression, accompanied by the phosphorylation of AKT (p-AKT) and ERK (p-ERK). However, it did not stimulate the phosphorylation of JNK and p38. SPP-86, LY29002, and U0126 significantly suppressed GDNF-induced claudin-5 and VE-cadherin expression, while SP600125 and SB203580 had almost no effect on GDNF-induced claudin-5 and VE-cadherin expression. GDNF-induced phosphorylation of AKT and ERK was also markedly attenuated by the anti-GDNF antibody (*Figure 3E*). Similarly, US-CM remarkably upregulated the expression of claudin-5, VE-cadherin, p-AKT, and p-ERK, which were also markedly reversed by SPP-86, LY29002, U0126, or anti-GDNF antibody (*Figure 3F–J*). These findings indicate that GDNF induces the expression of claudin-5 and VE-cadherin in hCMEC/D3 cells by activating both the PI3K/AKT and MAPK/ERK pathways.

## GDNF upregulated the claudin-5 expression in hCMEC/D3 cells by activating the PI3K/AKT/FOXO1 pathway

Claudin-5 is negatively regulated by the transcriptional repressor forkhead box O1 (FOXO1) (*Beard et al., 2020*). FOXO1 is also an important target of PI3K/AKT signaling. FOXO1 phosphorylation results in FOXO1 accumulation in the cytoplasm (*Zhang et al., 2011*) and lowers its level in the nucleus. Here, we investigated whether GDNF-induced claudin-5 expression is involved in FOXO1 nuclear exclusion. As shown in *Figure 4A*, both GDNF and US-CM significantly enhanced FOXO1 phosphorylation (p-FOXO1). Similarly, GDNF and US-CM increased the levels of phosphorylated and unphosphorylated FOXO1 in the cytoplasm and decreased the levels of nuclear FOXO1 (*Figure 4B*). Whether FOXO1 was involved in the GDNF-induced regulation of claudin-5 and VE-cadherin expression was investigated in hCMEC/D3 cells transfected with *FOXO1* small interfering RNA (siRNA). Silencing *FOXO1* significantly decreased FOXO1 levels in both whole-cell lysates and nucleus of hCMEC/D3 cells (*Figure 4C*), demonstrating *FOXO1* silencing efficacy. Consistent with our expectation, silencing *FOXO1* upregulated the expression of claudin-5 rather than VE-cadherin expression (*Figure 4D*). In contrast, high levels of FOXO1 were observed in both whole-cell lysates and nucleus of hCMEC/D3 cells that were transfected with plasmids containing *FOXO1*. Meanwhile, *FOXO1* overexpression resulted in a decrease in both basal and GDNF-induced claudin-5 expression (*Figure 4E*), consistent with the known role of FOXO1 on claudin-5 expression.

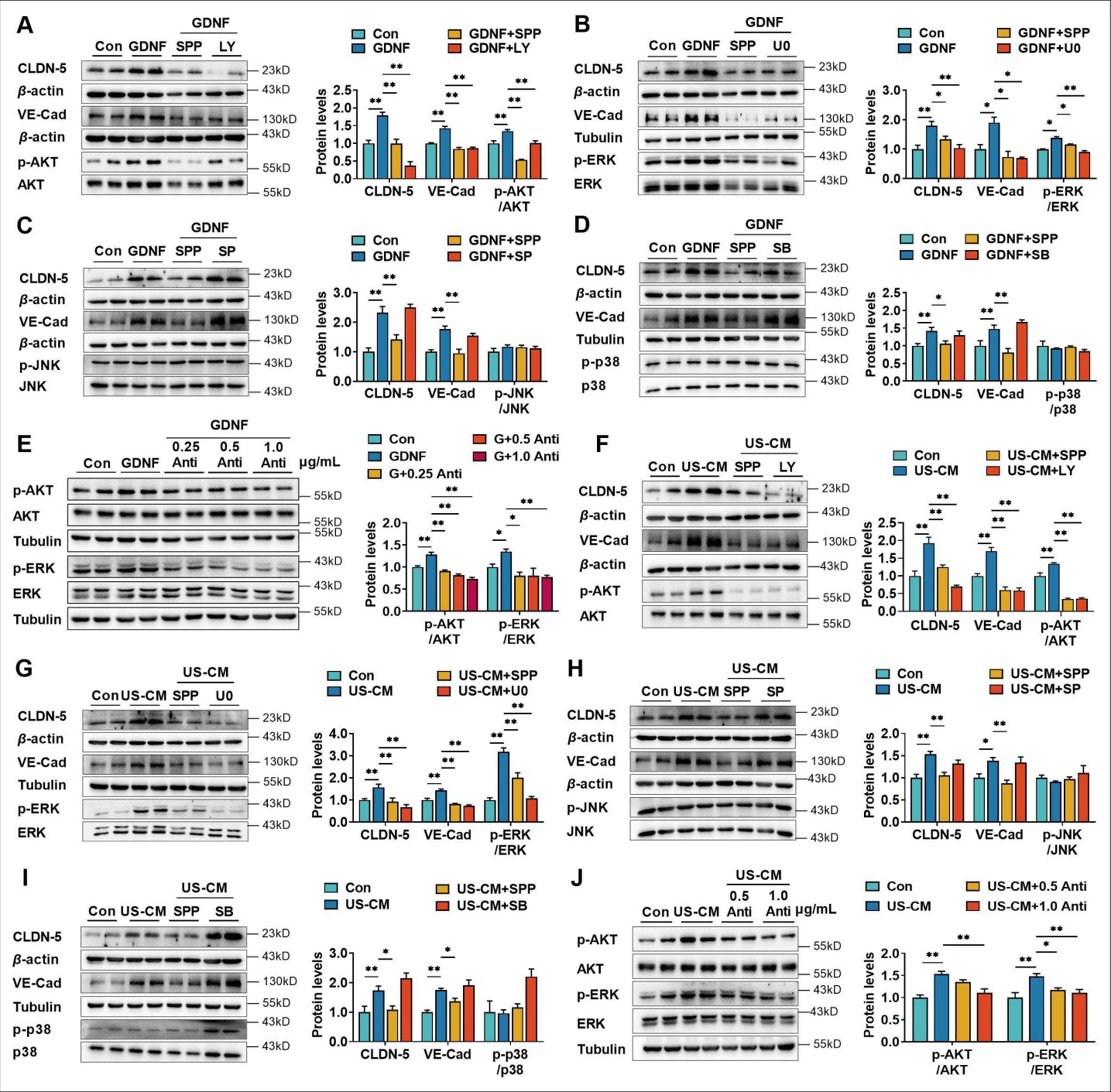

**Figure 3.** Glial cell line-derived neurotrophic factor (GDNF)-induced claudin-5 and VE-cadherin expression in hCMEC/D3 cells by activating the PI3K/ AKT and MAPK/ERK signaling. (**A**) Effects of 3 µM LY294002 (LY) on the levels of claudin-5, VE-cadherin, and p-AKT/AKT in hCMEC/D3 cells stimulated by 200 pg/ml GDNF. (**B**) Effects of 2 µM U0126 (U0) on the levels of claudin-5, VE-cadherin, and p-ERK/ERK in hCMEC/D3 cells stimulated by 200 pg/ ml GDNF. (**C**) Effects of 5 µM SP600125 (SP) on the levels of claudin-5, VE-cadherin, and p-JNK/JNK in hCMEC/D3 cells stimulated by 200 pg/ml GDNF. (**D**) Effects of 2 µM SB203580 (SB) on the levels of claudin-5, VE-cadherin, and p-p38/p38 in hCMEC/D3 cells stimulated by 200 pg/ml GDNF. (**E**) Effects of anti-GDNF antibody on the GDNF-induced p-AKT/AKT and p-ERK/ERK ratios. (**F**) Effects of 3 µM LY on the levels of claudin-5, VE-cadherin, and p-AKT/AKT in hCMEC/D3 cells stimulated by US-CM. (**G**) Effects of 2 µM U0 on the levels of claudin-5, VE-cadherin, and p-ERK/ERK in hCMEC/D3 cells stimulated by US-CM. (**H**) Effects of 5 µM SP on the levels of claudin-5, VE-cadherin, and p-JNK/JNK in hCMEC/D3 cells stimulated by US-CM. (**I**) Effects of 2 µM SB on the levels of claudin-5, VE-cadherin, and p-p38/p38 in hCMEC/D3 cells stimulated by US-CM. (**J**) Effects of anti-GDNF antibody on the US-CM-induced p-AKT/AKT and p-ERK/ERK ratios. The above data are shown as the mean ± SEM. Four biological replicates per group. One technical replicate for each biological replicate. *p < 0.05; **p < 0.01 by one-way ANOVA test followed by Fisher's LSD test or Welch's ANOVA test.

*Figure 3 continued on next page*

*Figure 3 continued*

The online version of this article includes the following source data for figure 3:

**Source data 1.** The western blot raw images in *Figure 3*.

**Source data 2.** The labeled western blot images in *Figure 3*.

**Source data 3.** Excel file containing summary data and data analysis of *Figure 3*.

Several reports have demonstrated that nuclear localization of FOXO1 is modulated by multiple pathways, including the PI3K/AKT (*Tang et al., 1999*) and MAPK/ERK (*Asada et al., 2007*) pathways. The effects of LY294002 and U0126 on GDNF-induced FOXO1 phosphorylation were measured in hCMEC/D3 cells. LY294002, but not U0126, significantly reversed the GDNF-induced alterations in total p-FOXO1, cytoplasmic p-FOXO1, cytoplasmic FOXO1, and nuclear FOXO1 (*Figure 4F*). Furthermore, LY294002 reversed the upregulation of claudin-5 expression induced by *FOXO1* siRNA and GDNF (*Figure 4G*).

It was reported that VE-cadherin also upregulates claudin-5 via inhibiting FOXO1 activities (*Taddei et al., 2008*). Effect of VE-cadherin on claudin-5 was studied in hCMEC/D3 cells silencing VE-cadherin. It was not consistent with Taddei et al. that silencing VE-cadherin only slightly decreased the mRNA level of claudin-5 without significant difference. Furthermore, basal and GDNF-induced claudin-5 protein levels were unaltered by silencing VE-cadherin (*Figure 4—figure supplement 1*). Thus, the roles of VE-cadherin in regulation of claudin-5 in BBB should be further investigated.

## GDNF upregulated VE-cadherin expression in hCMEC/D3 cells by activating the PI3K/AKT/ETS1 and MAPK/ERK/ETS1 signaling pathways

E26 oncogene homolog 1 (ETS1) is a transcription factor that binds to the ETS-binding site located in the proximal region of the VE-cadherin promoter, hence regulating the expression of VE-cadherin (*Lelièvre et al., 2000*; *Luo et al., 2022*). Activation of the PI3K/AKT (*He et al., 2023*; *Hui et al., 2018*) and MAPK/ERK (*Watanabe et al., 2004*) pathways was reported to upregulate the expression of ETS1. The previous results showed that GDNF-induced VE-cadherin and claudin-5 expression in hCMEC/D3 cells by activating the PI3K/AKT and MAPK/ERK pathways. Therefore, we hypothesized that GDNF modulated ETS1 levels to promote VE-cadherin and claudin-5 expression through PI3K/AKT and MAPK/ERK signaling pathways. Both US-CM and GDNF significantly increased total (*Figure 5A*) and nuclear ETS1 expression (*Figure 5B*). LY294002 and U0126 markedly attenuated GDNF-induced total (*Figure 5C*) and nuclear ETS1 expression (*Figure 5D*). To further confirm the involvement of the PI3K/AKT/ETS1 and MAPK/ERK/ETS1 pathways in GDNF-induced VE-cadherin expression, ETS1 in hCMEC/D3 cells was knocked down using *ETS1* siRNA. *ETS1* silencing saliently declined the expression levels of total (*Figure 5E*) and nuclear (*Figure 5F*) ETS1 in hCMEC/D3 cells, demonstrating silencing efficacy. In *ETS1* silencing hCMEC/D3 cells, GDNF no longer induced the expression of total (*Figure 5E*) and nuclear (*Figure 5F*) ETS1. Moreover, *ETS1* silencing substantially downregulated VE-cadherin expression and attenuated GDNF-induced VE-cadherin expression, while having minimal impact on both basal and GDNF-induced claudin-5 expression (*Figure 5G*).

## Brain GDNF deficiency increased BBB permeability partly due to the impairment of claudin-5 and VE-cadherin expression

To further demonstrate the positive effects of GDNF on BBB maintenance, GDNF in mice brains was knocked down via intracerebroventricular (*i.c.v*) injection of AAV-PHP.eB packaged with *Gdnf* short hairpin RNA (shRNA) (*Figure 6A*). Knockdown efficiency was confirmed through western blotting (*Figure 6B*). Consistent with *in vitro* results, GDNF knockdown greatly downregulated claudin-5 and VE-cadherin expression in the mice brains (*Figure 6B*). The integrity of BBB was assessed by examining the brain distributions of fluorescein and FITC-Dex. The results showed that specifically knocking down brain GDNF little affected plasma levels of fluorescein (*Figure 6C*) and FITC-Dex (*Figure 6F*), but significantly elevated the concentrations of fluorescein (*Figure 6D*) and FITC-Dex (*Figure 6G*) in the brains, leading to notable increases in the brain-to-plasma concentration ratios of two probes (*Figure 6E, H*). These alterations were consistent with the decline in claudin-5 and

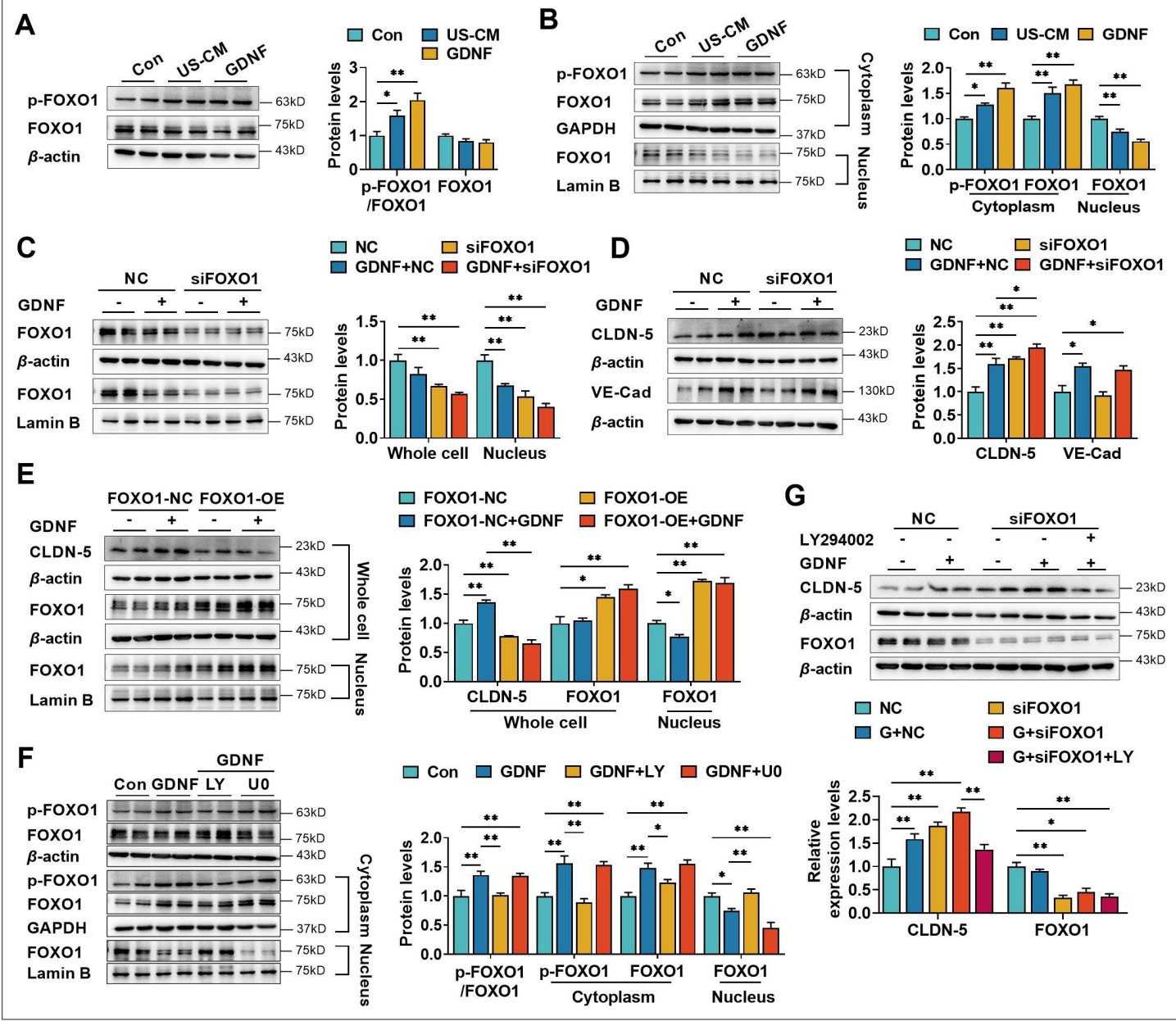

**Figure 4.** Glial cell line-derived neurotrophic factor (GDNF) induced the claudin-5 expression in hCMEC/D3 cells by activating the PI3K/AKT/FOXO1 pathway. Effects of US-CM and GDNF on the phosphorylated FOXO1 (p-FOXO1)/FOXO1 ratio, total FOXO1 expression (**A**), cytoplasmic p-FOXO1, cytoplasmic FOXO1, and nuclear FOXO1 expression (**B**). The expression levels of total and nuclear FOXO1 (**C**), claudin-5, and VE-cadherin (**D**) in hCMEC/D3 cells transfected with *FOXO1* siRNA (siFOXO1). NC: negative control. (**E**) Effects of *FOXO1* overexpression (FOXO1-OE) and GDNF on the expression levels of claudin-5, total FOXO1, and nuclear FOXO1. FOXO1-NC: negative control plasmids. (**F**) Effects of LY and U0 on GDNF-induced alterations of total p-FOXO1/FOXO1 ratio, cytoplasmic p-FOXO1, cytoplasmic FOXO1, and nuclear FOXO1 expression. (**G**) Effects of LY on the claudin-5 expression upregulated by siFOXO1. The above data are shown as the mean± SEM. Four biological replicates per group. One technical replicate for each biological replicate. *p < 0.05; **p < 0.01 by one-way ANOVA test followed by Fisher's LSD test, Welch's ANOVA test, or Kruskal–Wallis test.

The online version of this article includes the following source data and figure supplement(s) for figure 4:

**Source data 1.** The western blot raw images in *Figure 4*.

**Source data 2.** The labeled western blot images in *Figure 4*.

**Source data 3.** Excel file containing summary data and data analysis of *Figure 4*.

**Figure supplement 1.** The contribution of VE-cadherin on the glial cell line-derived neurotrophic factor (GDNF)-induced claudin-5 expression.

*Figure 4 continued on next page*

*Figure 4 continued*

**Figure supplement 1—source data 1.** The western blot raw images in *Figure 4—figure supplement 1*.

**Figure supplement 1—source data 2.** The labeled western blot images in *Figure 4—figure supplement 1*.

**Figure supplement 1—source data 3.** Excel file containing summary data and data analysis of *Figure 4—figure supplement 1*.

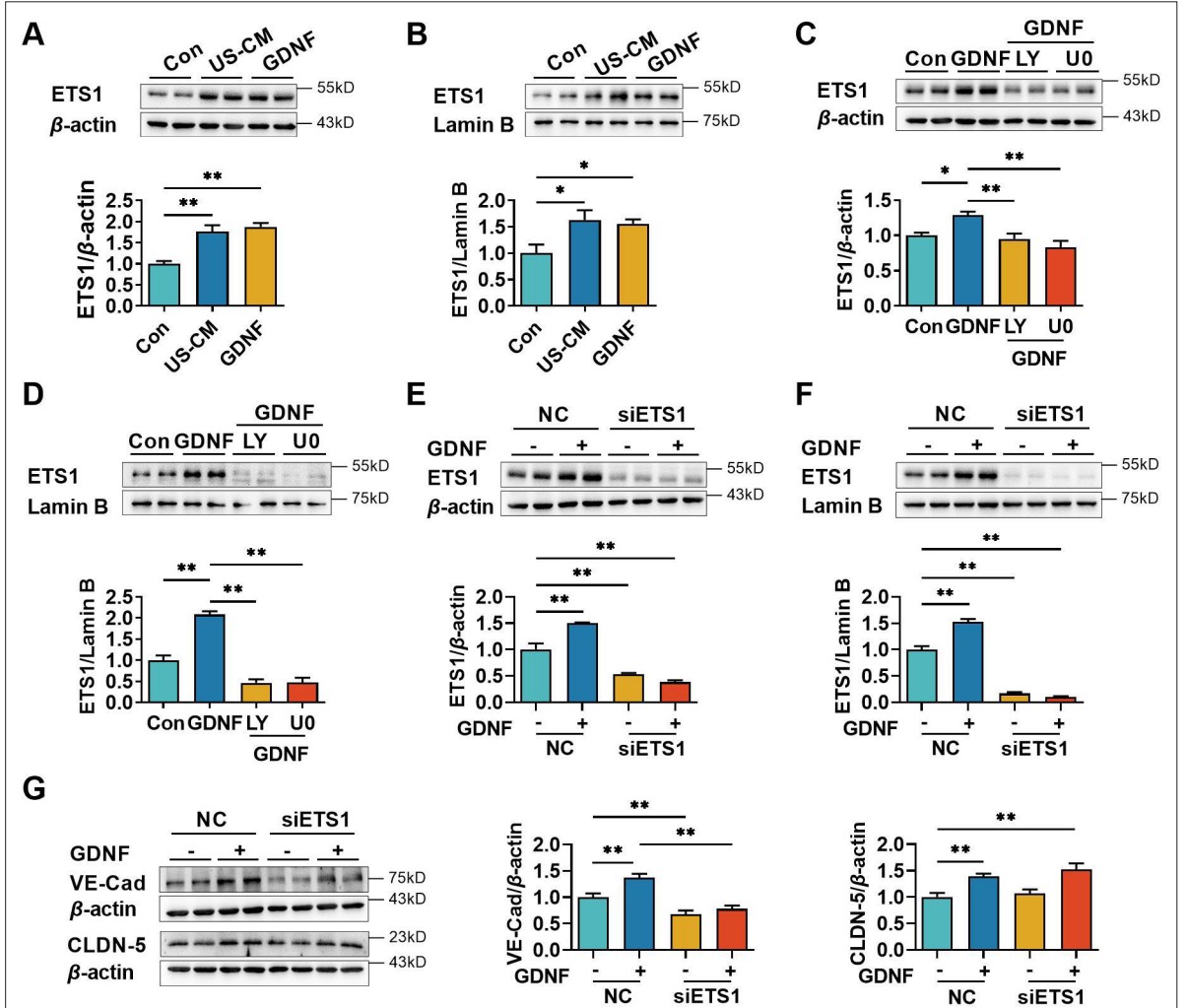

**Figure 5.** Glial cell line-derived neurotrophic factor (GDNF)-induced VE-cadherin expression in hCMEC/D3 cells by activating the PI3K/AKT/ETS1 and MAPK/ERK/ETS1 pathways. Effects of US-CM and GDNF on total (**A**) and nuclear (**B**) ETS1 expression. Effects of LY and U0 on 200 pg/ml GDNF-induced total (**C**) and nuclear (**D**) ETS1 expression. Expression levels of total (**E**) and the nuclear ETS1 (**F**) in hCMEC/D3 cells after knocking down *ETS1* with siRNA (siETS1). (**G**) Effects of GDNF and siETS1 on the expression of VE-cadherin and claudin-5. The above data are shown as the mean ± SEM. Four biological replicates per group. One technical replicate for each biological replicate. *p < 0.05; **p < 0.01 by one-way ANOVA test followed by Fisher's LSD test.

The online version of this article includes the following source data for figure 5:

**Source data 1.** The western blot raw images in *Figure 5*.

**Source data 2.** The labeled western blot images in *Figure 5*.

**Source data 3.** Excel file containing summary data and data analysis of *Figure 5*.

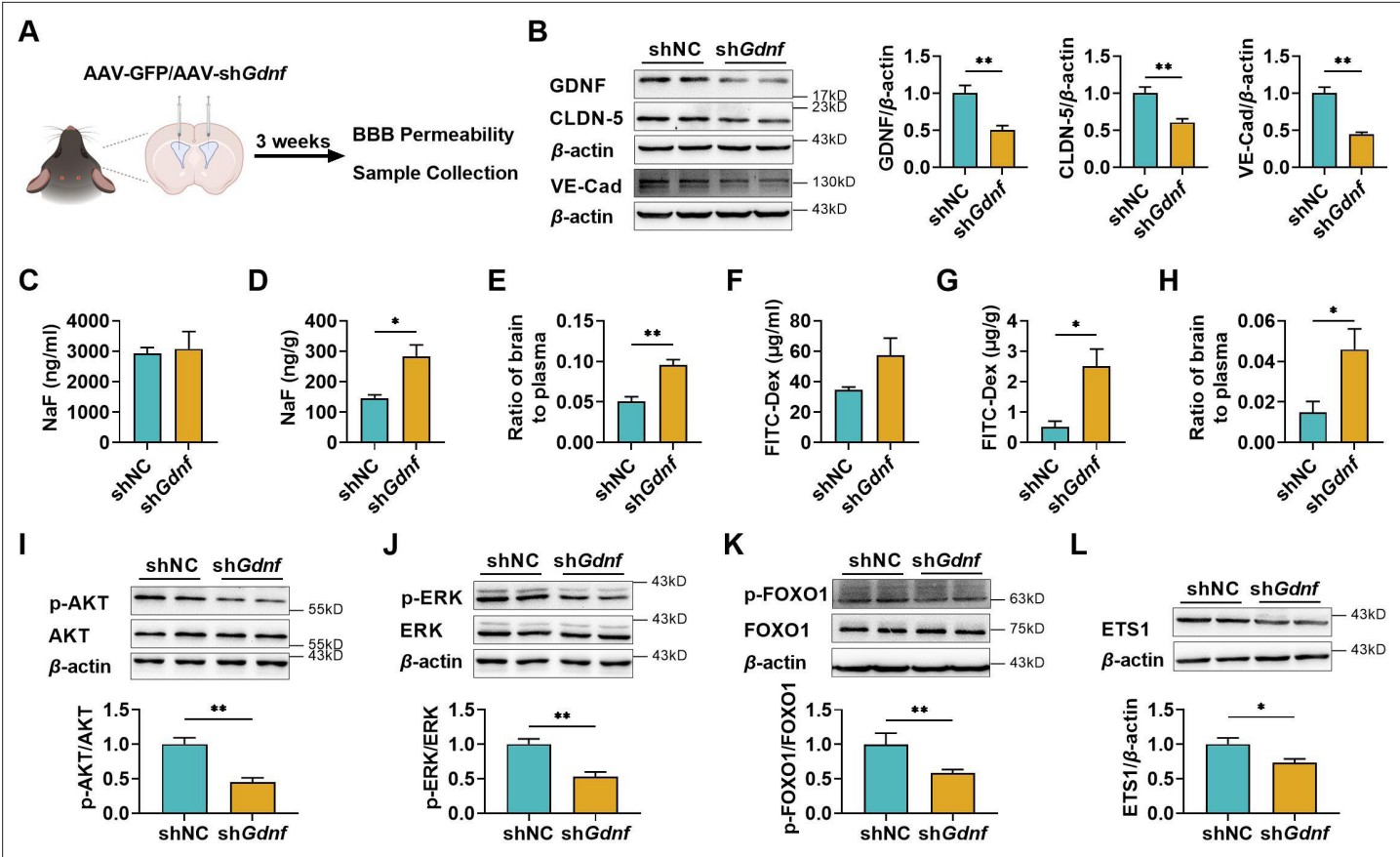

**Figure 6.** The deficiency of brain glial cell line-derived neurotrophic factor (GDNF) in mice increased the permeability of blood–brain barrier (BBB) and reduced claudin-5 and VE-cadherin expression in mice brains. (**A**) Experimental configuration of AAV-GFP (shNC) or AAV-sh*Gdnf* (sh*Gdnf*) intracerebroventricular injection. (**B**) Effects of brain-specific *Gdnf* silencing on the expression levels of GDNF, claudin-5, and VE-cadherin in the brains. Effects of brain-specific *Gdnf* silencing on NaF levels in plasma (**C**), brain (**D**), and the ratio of brain to plasma (**E**). Effects of brain-specific *Gdnf* silencing on FITC-Dex levels in plasma (**F**), brain (**G**), and the ratio of brain to plasma (**H**). The expression ratios of p-AKT/AKT (**I**), p-ERK/ERK (**J**), and p-FOXO1/FOXO1 (**K**) in the brains of *Gdnf* silencing mice. (**L**) The expression level of ETS1 in the brains of *Gdnf* silencing mice. The above data are shown as the mean ± SEM. Six biological replicates per group. One technical replicate for each biological replicate. *p < 0.05; **p < 0.01 by unpaired *t*-test, unpaired *t*-test with Welch's correction, or Mann–Whitney test.

The online version of this article includes the following source data for figure 6:

**Source data 1.** The western blot raw images in *Figure 6*.

**Source data 2.** The labeled western blot images in *Figure 6*.

**Source data 3.** Excel file containing summary data and data analysis of *Figure 6*.

VE-cadherin expression. In addition, significant reductions in the levels of p-AKT (*Figure 6I*), p-ERK (*Figure 6J*), p-FOXO1 (*Figure 6K*), and ETS1 expression (*Figure 6L*) were observed in the brains of *Gdnf* knockdown mice.

## The triple co-culture BBB model better predicted the permeabilities of drugs across BBB

In this study, 18 drugs were utilized to further investigate the superiority of the triple co-culture BBB model over the hCMEC/D3 mono-culture BBB model. The $P_{app}$ of 18 drugs from the apical to the basolateral side based on the hCMEC/D3 mono-culture ($P_{app, Mono}$) and triple co-culture ($P_{app, Triple}$) BBB models are measured and listed in *Table 1*. The results showed that $P_{app, Triple}$ values of all tested drugs were lower than the $P_{app, Mono}$ values. Significant differences were observed in 14 out of 18 drugs. The predicted permeability coefficient-surface area product values ($PS$) of the tested drugs were, respectively, calculated based on their $P_{app, Mono}$ values ($PS_{Pre, Mono}$) and $P_{app, Triple}$ values ($PS_{Pre, Triple}$). The predicted $PS$ values were further compared to their corresponding observations ($PS_{obs}$). The results showed that

**Table 1.** The unbound fraction in brain ($f_{u, brain}$), the observed $PS_{Obs}$, and the predicted $PS$ ($PS_{Pre}$), $P_{app}$ across the hCMEC/D3 mono-culture model ($P_{app, Mono}$) and triple co-culture model ($P_{app, Triple}$) of the tested drugs.

| Compounds | $f_{u, brain}$ | $PS_{Obs}$ µl/min/g | $P_{app, Mono}$ cm/s × 10⁻⁶ | $PS_{Pre, Mono}$ µl/min/g | $P_{app, Triple}$ cm/s × 10⁻⁶ | $PS_{Pre, Triple}$ µl/min/g |
|---|---|---|---|---|---|---|
| Amantadine | 0.1985[*] | 116.10[†] | 6.84 ± 0.95 | 310.22 | 3.64 ± 0.26 | 165.23 |
| Amitriptyline | 0.01[‡] | 4608.00[***] | 15.24 ± 0.64 | 13,716.00 | 14.61 ± 0.27 | 13,149.00 |
| Bupropion | 0.12[§] | 1519.20[†] | 15.19 ± 0.20 | 1139.58 | 11.34 ± 0.44 | 850.58 |
| Carbamazepine | 0.116[¶] | 959.40[†] | 34.37 ± 1.26 | 2666.89 | 11.71 ± 0.15 | 908.69 |
| Clozapine | 0.014[**] | 2260.80[†] | 38.97 ± 0.54 | 25,052.57 | 12.87 ± 2.06 | 8272.72 |
| Donepezil | 0.07[††] | 1581.30[†] | 20.91 ± 0.75 | 2688.43 | 14.47 ± 0.84 | 1860.43 |
| Doxepin | 0.025[†] | 2192.40[†] | 16.88 ± 1.08 | 6076.80 | 10.66 ± 0.92 | 3837.60 |
| Fluoxetine | 0.004[†] | 2698.20[†] | 11.48 ± 0.85 | 25,830.00 | 9.97 ± 1.03 | 22,430.25 |
| Gabapentin | 0.782[†] | 162.90[†] | 16.75 ± 1.62 | 192.77 | 8.78 ± 0.23 | 101.05 |
| Lamotrigine | 0.273[†] | 126.00[†] | 14.26 ± 0.37 | 470.11 | 5.97 ± 0.11 | 196.88 |
| Metoclopramide | 0.365[†] | 125.10[†] | 14.14 ± 1.44 | 348.66 | 6.63 ± 0.42 | 163.41 |
| Midazolam | 0.045[‡‡] | 2727.00[†] | 25.30 ± 1.00 | 5060.00 | 19.09 ± 0.24 | 3818.00 |
| Mirtazapine | 0.08[†] | 1912.50[†] | 23.44 ± 0.44 | 2637.00 | 17.86 ± 0.21 | 2009.25 |
| Olanzapine | 0.034[†] | 2279.70[†] | 22.91 ± 3.80 | 6064.41 | 12.49 ± 0.53 | 3306.18 |
| Prazosin | 0.09[§§] | 169.20[¶¶] | 5.61 ± 0.38 | 560.93 | 2.81 ± 0.52 | 280.99 |
| Risperidone | 0.099[†] | 849.60[†] | 16.10 ± 2.87 | 1463.64 | 11.70 ± 0.25 | 1063.64 |
| Venlafaxine | 0.205[*] | 584.10[†] | 9.58 ± 0.28 | 420.60 | 8.25 ± 0.36 | 362.02 |
| Verapamil | 0.033[‡] | 335.70[‡] | 7.21 ± 0.41 | 1965.24 | 5.56 ± 0.06 | 1517.67 |

One technical replicate of four biological replicates per group.

[*](***Esaki et al., 2019***).
[†](***Summerfield et al., 2007***).
[‡](***Fridén et al., 2011***).
[§](***Bhattacharya et al., 2021***).
[¶](***Maurer et al., 2005***).
[**](***Cremers et al., 2012***).
[††](***Di et al., 2011***).
[‡‡](***Kodaira et al., 2011***).
[§§](***Zhou et al., 2009***).
[¶¶](***Di Marco et al., 2019***).
[***](***Avdeef and Sun, 2011***).

The online version of this article includes the following source data for table 1:

**Source data 1.** The apparent permeability coefficients of 18 tested drugs from mono- or triple- culture blood–brain barrier (BBB) model.

the predictive accuracy of $PS_{Pre, Triple}$ was superior to $PS_{Pre, Mono}$. Except for verapamil, amitriptyline, fluoxetine, and clozapine, the predicted $PS_{Pre, Triple}$ values of the other 14 drugs in the triple co-culture BBB models were within the 0.5- to 2-folds of $PS_{obs}$ (***Figure 7B***). However, in the hCMEC/D3 mono-culture BBB model, only seven predicted $PS_{Pre, Mono}$ values were within the 0.5- to 2-folds range of their observations (***Figure 7A***).

## Discussion

The main findings of the study were to successfully develop an *in vitro* triple co-culture BBB model consisting of hCMEC/D3, U251, and SH-SY5Y cells and to confirm the involvement of neurons in BBB

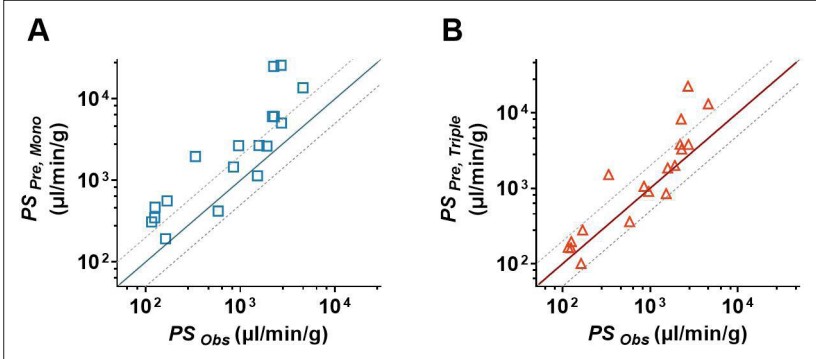

**Figure 7.** *In vitro*/*in vivo* correlation assay of blood–brain barrier (BBB) permeability. (**A**) The comparison of the estimated permeability coefficient-surface area product ($PS_{Pre,\ Mono}$) recalculated from $P_{app,\ Mono}$ with the observed *in vivo* PS values ($PS_{Obs}$). (**B**) The comparison of the estimated permeability coefficient-surface area product ($PS_{Pre,\ Triple}$) recalculated from $P_{app,\ Triple}$ with the observed *in vivo* PS values ($PS_{Obs}$). The solid line represents a perfect prediction, and the dashed lines represent the 0.5- to 2-folds of their observations. The $PS_{Obs}$ values were determined by *in situ* brain perfusion in rodents, which were collected from the literature.

maintenance as well as the possible mechanisms. Co-culture with U251 and/or SH-SY5Y cells markedly promoted the TEER of hCMEC/D3 cells and reduced the leakage of fluorescein and FITC-Dex due to the upregulation of claudin-5 and VE-cadherin expression.

The roles of claudin-5 and VE-cadherin in the maintenance of BBB function have been demonstrated (*Dejana et al., 2008*; *Hashimoto et al., 2023*; *Li et al., 2018*; *Ohtsuki et al., 2007*). It was reported that *Cld-5*-deficient mice exhibited BBB impairment, allowing the transport of small molecules (<800 D) across BBB (*Nitta et al., 2003*). In contrast, claudin-5 overexpression significantly restricted the permeability of inulin across the conditionally immortalized rat brain capillary endothelial cell monolayer (*Ohtsuki et al., 2007*). Moreover, the claudin-5 expression in hCMEC/D3 is much lower than in human brain microvessels (*Eigenmann et al., 2013*; *Ohtsuki et al., 2013*; *Weksler et al., 2013*), which may cause the low TEER values. VE-cadherin, a major member of the cadherin family in endothelial cells, is also required for BBB integrity (*Dejana et al., 2008*; *Li et al., 2018*). The absence of VE-cadherin resulted in faulty cell-to-cell junctions and disrupted distribution of ZO-1 (*Sauteur et al., 2017*; *Tunggal et al., 2005*). Strongly negative correlations between $P_{app}$ values of fluorescein or FITC-Dex and expression of claudin-5 or VE-cadherin further demonstrated the importance of claudin-5 and VE-cadherin in BBB integrity.

Neurons and astrocytes, as important components of the NVU, may be involved in the formation and maintenance of BBB function. Several reports showed that co-culture with astrocytes CC-2565 or SC-1810 enhanced the TEER of hCMEC/D3 cells (*Hatherell et al., 2011*), whereas co-culture of RBE4.B cells with primary rat neurons and astrocytes showed a lower permeability of [³H] sucrose than mono-culture of RBE4.B cells (*Schiera et al., 2005*). Similarly, co-cultured with hCMEC/D3 cells and 1321N1 (astrocytes) or 1321N1+SH-SY5Y cells possess higher TEER values and lower permeability to Lucifer yellow than hCMEC/D3 alone. Compared with double co-culture of hCMEC/D3 and 321N1 cells, the triple co-culture of hCMEC/D3, 1321N1, and SH-SY5Y cells showed higher TEER values (*Barberio et al., 2022*). Our study also demonstrated that co-culture with U251 and/or SH-SY5Y cells significantly lowered BBB permeability and upregulated VE-cadherin or claudin-5 expression in hCMEC/D3 cells.

Next, we focused on the molecular mechanisms by which neurons and possibly astrocytes upregulated the VE-cadherin and claudin-5 expression in BMECs. Co-culture with SH-SY5Y cells significantly upregulated claudin-5 and VE-cadherin expression in hCMEC/D3 cells. In the double co-culture with SH-SY5Y cells or triple co-culture BBB models, hCMEC/D3 cells were not in direct contact with SH-SY5Y cells, indicating that the interaction between SH-SY5Y and hCMEC/D3 cells depended on the release of some active compounds. It was consistent with the above deduction that the S-CM also markedly induced claudin-5 and VE-cadherin expression. Different from S-CM, U-CM mainly upregulated the VE-cadherin expression and just had a slight impact on claudin-5. In general, neurons but also astrocytes secrete some neurotrophic factors (*Lonka-Nevalaita et al., 2010*; *Sweeney et al.,*

*2019*) that could contribute to the maintenance of structural stability of BBB. High levels of bFGF, GDNF, IGF-1, and TGF-$\beta$ were detected in U-CM, S-CM, and US-CM. Further research showed that only GDNF-induced VE-cadherin and claudin-5 expression in hCMEC/D3 cells in a concentration-dependent manner, and the anti-GDNF antibody attenuated claudin-5 and VE-cadherin expression induced by US-CM or GDNF. These results indicated that neurons upregulated claudin-5 and VE-cadherin expression through GDNF secretion. Levels of GDNF in S-CM were higher than those in U-CM, which seemed to partly explain why S-CM has a stronger promoting effect on claudin-5 than U-CM. The roles of GDNF in BBB maintenance and the regulation of claudin-5 and VE-cadherin expression were further confirmed using brain-specific *Gdnf* knockdown C57BL/6J mice. Consistent with our *in vitro* results, brain-specific *Gdnf* silencing greatly increased BBB penetration of fluorescein and FITC-Dex, accompanied by the downregulation of claudin-5 and VE-cadherin expression.

GDNF is mainly expressed in astrocytes and neurons (*Lonka-Nevalaita et al., 2010*; *Pochon et al., 1997*). In adult animals, GDNF is mainly secreted by striatal neurons rather than astrocytes and microglial cells (*Hidalgo-Figueroa et al., 2012*). The present study also shows that GDNF mRNA levels in SH-SY5Y cells were significantly higher than that in U251 cells. GDNF was also detected in CM from SH-SY5Y cells. All these results demonstrate that neurons may secrete GDNF.

Generally, GDNF activates several signal transduction pathways, such as the PI3K/AKT and MAPK signaling pathways (*Fielder et al., 2018*) by forming a heterohexameric complex with two GFR$\alpha$ molecules and RET receptors. It was also reported that GDNF improved BBB barrier function due to the activation of MAPK/ERK1 (*Dong and Ubogu, 2018*) and PI3K/AKT (*Liu et al., 2022*) signaling. Consistent with previous reports, we found that signaling inhibitors SPP-86, PP2, LY294002, and U0126 markedly attenuated US-CM- and GDNF-induced claudin-5 and VE-cadherin expression in hCMEC/D3 cells, inferring that GDNF promoted claudin-5 and VE-cadherin expression via activating both the PI3K/AKT and MAPK/ERK pathways.

Signal transduction pathways control gene expression by modifying the function of nuclear transcription factors. The nuclear accumulation of FOXO1 negatively regulates claudin-5 expression (*Beard et al., 2020*; *Taddei et al., 2008*). FOXO1 is an important target of the PI3K/AKT signaling axis (*Zhang et al., 2011*), and AKT-induced phosphorylation of FOXO1 results in cytoplasmic FOXO1 accumulation and decreases nuclear FOXO1 accumulation (*Zhang et al., 2011*). From these results, we inferred that GDNF-induced claudin-5 expression in hCMEC/D3 cells may be involved in the activation of PI3K/AKT/FOXO1 pathway. Similarly, both US-CM and GDNF increased cytoplasmic p-FOXO1 and decreased nuclear FOXO1 in hCMEC/D3 cells, which was reversed by LY29002 rather than U0126. Roles of FOXO1 in GDNF-induced claudin-5 were verified through silencing and overexpressing *FOXO1* in hCMEC/D3 cells. *FOXO1* silencing enhanced claudin-5 but not VE-cadherin expression, accompanied by a decline in total and nuclear FOXO1. In contrast, *FOXO1* overexpression significantly decreased claudin-5 expression. In hCMEC/D3 cells overexpressing *FOXO1*, GDNF lost its promotion effect on claudin-5 expression. It was noticed that U0126 attenuated the GDNF-induced upregulation of claudin-5 but had minimal impact on GDNF-mediated FOXO1 phosphorylation and the decline of nuclear FOXO1. In Sertoli cells, it was found that testosterone-stimulated claudin-5 expression by activating the RAS/RAF/ERK/CREB pathway (*Bulldan et al., 2016*). The transcriptional regulation of claudin-5 by CREB was confirmed in bEnd.3 (mouse brain endothelial cell). CREB overexpression significantly increased both gene and protein expression of claudin-5. In contrast, depletion of CREB decreased claudin-5 expression in gene and protein levels (*Li et al., 2022a*). However, another report showed that in human lung microvascular endothelial cells, U0126 attenuated phosphorylation of ERK and lipopolysaccharide-stimulated claudin-5 damage, indicating activation of MAPK/ERK pathway impaired rather than promoted claudin-5 expression (*Liu et al., 2019*). Thus, the real mechanisms that GDNF-induced activation of the RET/MAPK/ERK pathway promotes claudin-5 expression need further investigation.

ETS1 is a member of the ETS family that plays an important role in cell adhesion, migration, and blood vessel information. ETS1 binds to an ETS-binding site located in the proximal region of the VE-cadherin promoter, controlling VE-cadherin expression (*Lelièvre, 2001*). Several studies have demonstrated the role of ETS1 in the regulation of VE-cadherin expression. For example, IFN-$\gamma$ and TNF-$\alpha$ impaired BBB integrity by decreasing ETS1-induced VE-cadherin expression (*Luo et al., 2022*). *ETS1* silencing reduced VE-cadherin expression in umbilical vein endothelial cells (*Colás-Algora et al., 2020*). In contrast, *ETS1* overexpression induced VE-cadherin expression in mouse brain capillary

endothelial cells and fibroblasts (*Lelièvre et al., 2000*). As expected, *ETS1* silencing resulted in a decrease in the expression of VE-cadherin in hCMEC/D3 cells, but claudin-5 expression remained unaffected. Additionally, *ETS1* silencing removed the inductive effect of GDNF on VE-cadherin expression while unaffecting the upregulation of claudin-5 induced by GDNF. The activation of the PI3K/AKT (*He et al., 2023*; *Hui et al., 2018*) and MAPK/ERK (*Watanabe et al., 2004*) pathways promotes ETS1 expression. In our findings, US-CM and GDNF significantly increased total and nuclear ETS1 levels, which were eliminated by signaling inhibitors LY294002 and U0126. Both our results and previous research provide evidence that GDNF upregulates ETS1 expression via the activation of PI3K/AKT and MAPK/ERK signaling to promote VE-cadherin expression.

Claudin-5 expression is also regulated by VE-cadherin (*Taddei et al., 2008*). Differing from the previous reports, silencing VE-cadherin with siRNA only slightly affected basal and GDNF-induced claudin-5 expression. The discrepancies may come from different characteristics of the tested cells. Several reports have supported the above deduction. In retinal endothelial cells, hyperglycemia remarkably reduced claudin-5 expression (but not VE-cadherin) (*Saker et al., 2014*). However, in hCMEC/D3 cells, hypoglycemia significantly decreased claudin-5 expression but hyperglycemia increased VE-cadherin expression (*Sajja et al., 2014*).

The present study showed that characteristics of *in vitro* triple co-culture BBB model were superior to those of hCMEC/D3 mono-culture BBB model. The hCMEC/D3 mono-culture and triple co-culture BBB models were used to try to predict the *PS* values of 18 drugs by comparing them with their observations. The prediction success rate (14/18) of triple co-culture BBB model was greater than that of hCMEC/D3 mono-culture BBB model (7/18). However, poor predictions were observed for verapamil, amitriptyline, fluoxetine, and clozapine, which may partly be due to inaccuracies in their $f_{u, brain}$ values. These four drugs are high protein-binding drugs. Due to the methodological discordance and limitations of historic devices for these drugs, the $f_u \leq 0.01$ maybe with low confidence and

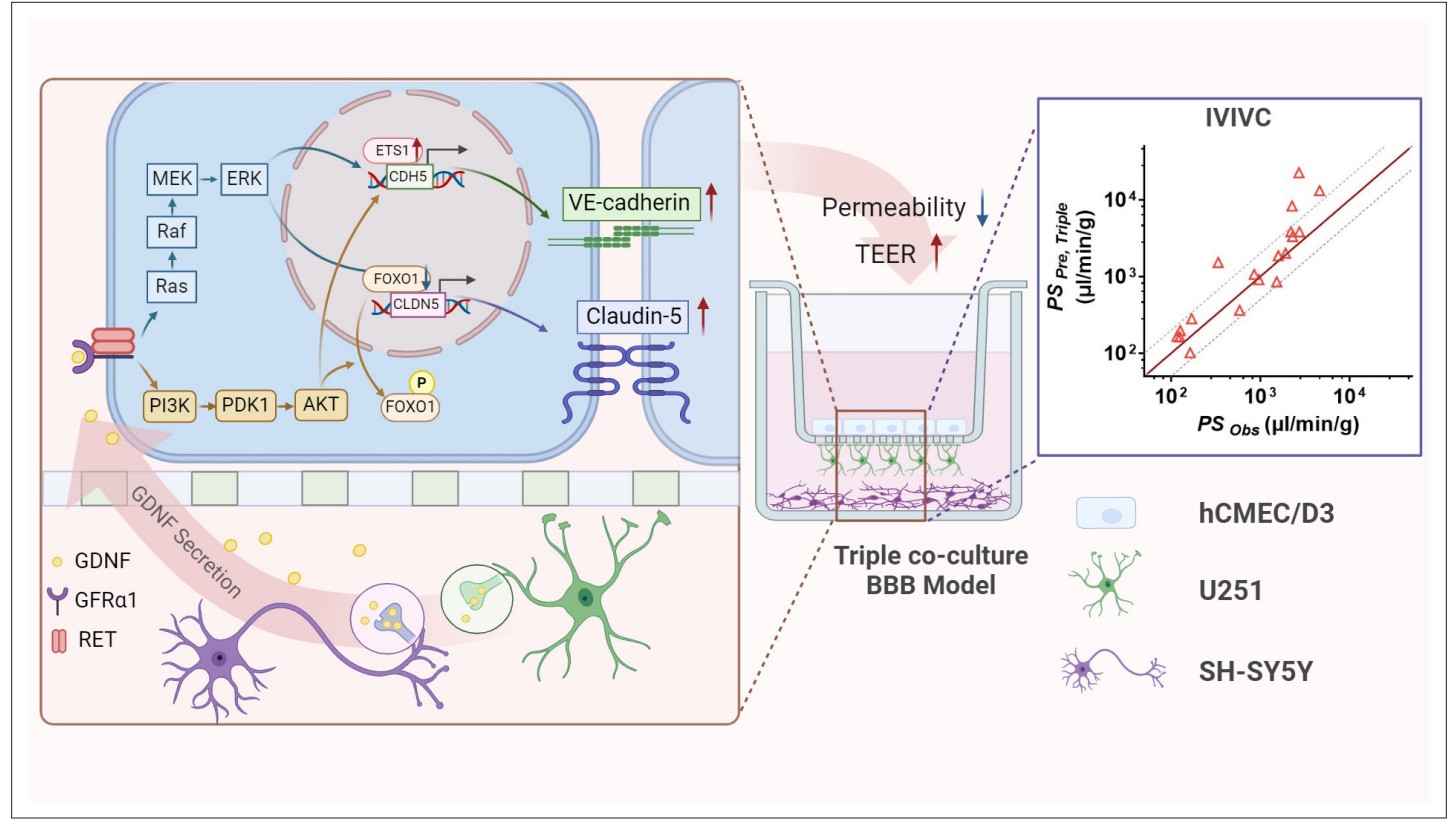

**Figure 8.** The mechanism of neurons and astrocytes induced the integrity of brain endothelial cells. Neurons but also astrocytes trigger the activation of PI3K/AKT and MAPK/ERK pathways in brain endothelial cells by glial cell line-derived neurotrophic factor (GDNF) secretion, which in turn regulates transcription factors of claudin-5 (FOXO1) and VE-cadherin (ETS1) to promote claudin-5 and VE-cadherin expression and leads to the enhancement of blood–brain barrier (BBB) integrity. Meanwhile, with the increase in barrier integrity, the *in vitro* BBB model also obtained a stronger *in vivo* correlation.

accuracy (*Bowman and Benet, 2018*; *Di et al., 2017*). For example, $f_{u, brain}$ values of amitriptyline from different reports display large differences [0.002 (*Sánchez-Dengra et al., 2021*), 0.0071 (*Weber et al., 2013*)]. The same large differences of $f_{u, brain}$ values were also found in fluoxetine [0.0023 (*Maurer et al., 2005*), 0.004 (*Summerfield et al., 2007*), 0.00094 (*Liu et al., 2005*)], and clozapine [0.0056 (*Bhyrapuneni et al., 2018*), 0.014 (*Cremers et al., 2012*), 0.011 (*Summerfield et al., 2007*)].

In summary, our triple co-culture BBB model outperformed the mono-culture or double co-culture BBB models, mainly attributing to the fact that neurons and possibly astrocytes upregulated claudin-5 and VE-cadherin expression by secreting GDNF through activating PI3K/AKT and MAPK/ERK pathways. Additionally, the developed *in vitro* triple co-culture BBB model accurately predicted the *in vivo* BBB permeability of CNS drugs. This suggests the potential of our triple co-culture BBB model for utilization in CNS candidate screening during the drug development process (*Figure 8*).

However, the study also has some limitations. In addition to neurons and astrocytes, other cells such as microglia, pericytes, and vascular smooth muscle cells, especially pericytes, may also affect BBB function. How pericytes affect BBB function and interaction among neurons, astrocytes, and pericytes needs further investigation.

# Materials and methods

**Key resources table**

| Reagent type (species) or resource | Designation | Source or reference | Identifiers | Additional information |
|---|---|---|---|---|
| Antibody | β-Actin (mouse monoclonal) | Proteintech | 66009 RRID:AB_2883475 | 1:10,000 |
| Antibody | GAPDH (mouse monoclonal) | Absin | Abs830030ss RRID:AB_2811228 | 1:50,000 |
| Antibody | β-Tubulin (mouse monoclonal) | Fdbio Science | FD0064 RRID:AB_3076327 | 1:10,000 |
| Antibody | Lamin B (mouse monoclonal) | Proteintech | 66095 RRID:AB_2721256 | 1:10,000 |
| Antibody | Claudin-5 (rabbit polyclonal) | Wanleibio | WL03731 RRID:AB_3076320 | 1:1000 |
| Antibody | Occludin (rabbit polyclonal) | Wanleibio | WL01996 RRID:AB_3076325 | 1:500 |
| Antibody | ZO-1 (mouse polyclonal) | Proteintech | 21773-1-AP RRID:AB_10733242 | 1:5000 |
| Antibody | VE-cadherin (rabbit polyclonal) | Wanleibio | WL02033 RRID:AB_3076321 | 1:1000 |
| Antibody | β-Catenin (rabbit polyclonal) | Wanleibio | WL0962a RRID:AB_3076323 | 1:5000 |
| Antibody | BCPR (rabbit polyclonal) | CST | 4477S RRID:AB_10544928 | 1:1000 |
| Antibody | P-gp (rabbit monoclonal) | CST | 13978S RRID:AB_2798357 | 1:1500 |
| Antibody | p-AKT (mouse monoclonal) | Huaan Biotechnology | ET1607 RRID:AB_2940863 | 1:2000 |
| Antibody | AKT (mouse monoclonal) | Huaan Biotechnology | ET1609 RRID:AB_3069857 | 1:2000 |
| Antibody | p-ERK (rabbit polyclonal) | Proteintech | 28733-1-AP RRID:AB_2881202 | 1:1000 |
| Antibody | ERK (rabbit polyclonal) | Proteintech | 11257-1-AP RRID:AB_2139822 | 1:1000 |
| Antibody | p-p38 (rabbit monoclonal) | CST | 4511S RRID:AB_10890701 | 1:250 |

*Continued on next page*

*Continued*

| Reagent type (species) or resource | Designation | Source or reference | Identifiers | Additional information |
|---|---|---|---|---|
| Antibody | p38 (rabbit monoclonal) | CST | 8690S RRID:AB_10999090 | 1:250 |
| Antibody | p-JNK (rabbit polyclonal) | Wanleibio | WL01813 RRID:AB_2910628 | 1:1000 |
| Antibody | JNK (rabbit polyclonal) | Wanleibio | WL01295 RRID:AB_3064853 | 1:1000 |
| Antibody | FOXO1 (rabbit polyclonal) | Proteintech | 18592 RRID:AB_2934932 | 1:1000 |
| Antibody | p-FOXO1 (rabbit polyclonal) | Wanleibio | WL03634 RRID: AB_3076326 | 1:1000 |
| Antibody | ETS1 (mouse monoclonal) | Santa Cruz | sc-55581 RRID:AB_831289 | 1:500 |
| Antibody | ETS1 (mouse monoclonal) | Proteintech | 66598 RRID:AB_2881958 | 1:3000 |
| Cell line (*Homo sapiens*) | hCMEC/D3 cells | JENNIO Biological Technology, Guangzhou, China | Cat#JNO-H0520 RRID:CVCL_U985 | Authenticated (STR profiling) |
| Cell line (*H. sapiens*) | U251 cells | Cellcook Biological Technology, Guangzhou, China | Cat#CC1701 RRID:CVCL_0021 | Authenticated (STR profiling) |
| Cell line (*H. sapiens*) | SH-SY5Y cells | Cellcook Biological Technology, Guangzhou, China | Cat#CC2101 RRID:CVCL_0019 | Authenticated (STR profiling) |
| Software, algorithm | GraphPad Prism | Version 8.0.2 | RRID:SCR_002798 | |
| Software, algorithm | BioTek Cytation 5 Cell Imaging Multi-Mode Reader | BioTek Cytation 5 | RRID:SCR_019732 | |
| Software, algorithm | QuantStudio 3 Real Time PCR System | QuantStudio 3 | RRID:SCR_018712 | |
| Software, algorithm | FACS Celesta Flow Cytometer | BD Biosciences | RRID:SCR_019597 | |
| Software, algorithm | Flowjo software | Version 10.4 | RRID:SCR_008520 | |
| Commercial assay or kit | GDNF-Elisa kit | R&D system RRID:SCR_006140 | Cat#212-GD | |
| Commercial assay or kit | bFGF-Elisa kit | Elabscience RRID:SCR_025982 | Cat#E-EL-H6042 | |
| Commercial assay or kit | IGF-1-Elisa kit | Elabscience RRID:SCR_025982 | Cat#E-EL-H0086 | |
| Commercial assay or kit | TGF-$\beta$-Elisa kit | Elabscience RRID:SCR_025982 | Cat#E-EL-0162 | |
| Peptide, recombinant protein | GDNF | R&D system RRID:SCR_006140 | Cat#212-GD | |
| Peptide, recombinant protein | bFGF | MedChemExpress RRID:SCR_025062 | Cat#HY-P7331 | |
| Peptide, recombinant protein | IGF-1 | MedChemExpress RRID:SCR_025062 | Cat#HY-P70783 | |
| Peptide, recombinant protein | TGF-$\beta$ | MedChemExpress RRID:SCR_025062 | Cat#HY-P70543 | |
| Chemical compound, drug | SPP-86 | MedChemExpress RRID:SCR_025062 | Cat#HY-110193 | |
| Chemical compound, drug | PP2 | MedChemExpress RRID:SCR_025062 | Cat#HY-13805 | |

Continued

| Reagent type (species) or resource | Designation | Source or reference | Identifiers | Additional information |
|---|---|---|---|---|
| Chemical compound, drug | LY294002 | MedChemExpress RRID:SCR_025062 | Cat#HY-10108 | |
| Chemical compound, drug | U0126 | MedChemExpress RRID:SCR_025062 | Cat#S1102 | |
| Chemical compound, drug | SP600125 | Selleck RRID:SCR_003823 | Cat#HY-12041 | |
| Chemical compound, drug | SB203580 | MedChemExpress RRID:SCR_025062 | Cat#HY-10256 | |

## Cell culture and viability assay

Rat BMECs were isolated from Sprague-Dawley rats (male, 7–10 days old, Sino-British Sippr/BKLaboratory Animal Ltd, Shanghai, China) as the described method (*Ji et al., 2013*; *Li et al., 2013*) and cultured in Dulbecco's Modified Eagle Media (DMEM)/F12 (#12500-039, Gibco, Carlsbad, CA, USA) containing 10% fetal bovine serum (FBS) (#10100147C, Gibco, Carlsbad, CA, USA) and 62.5 µg/ml penicillin and 100 µg/ml streptomycin (SunShine Biotechnology Co., Ltd, Nanjing, China). Then, hCMEC/D3, U251, and SH-SY5Y cells were cultured in DMEM/F12 containing 10% FBS, 62.5 µg/ml penicillin and 100 µg/ml streptomycin. Cell viability was assessed using a CCK-8 kit (Beyotime Biotechnology, Shanghai, China), and the results were expressed as the fold of control.

## Establishment of the triple co-culture model

Although hCMEC/D3 cells have poor barrier properties and low TEER compared to human physiological BBB, the use of human BMECs may be restricted by the acquisition of materials and ethical approval. Isolation and purification of primary BMECs are time-consuming and laborious. Moreover, culture conditions can alter transcriptional activity (*Qi et al., 2023*). All limit the establishment of BBB models based on primary human BMECs for high-throughput screening. Here, hCMEC/D3 cells were selected to establish an *in vitro* BBB model.

The establishment process of the triple co-culture model is illustrated in *Figure 9*. SH-SY5Y cells were seeded at a density of $4.5 \times 10^4$ cells/cm$^2$ in plates and differentiated with 10 µM retinoic acid (Sigma-Aldrich, St. Louis, MO, USA) for 72 hr. The differentiated SH-SY5Y cells were cultured in the fresh DMEM/F12 medium containing 10% FBS. U251 cells were seeded at $2 \times 10^4$ cells/cm$^2$ on the bottom of Transwell inserts (PET, 0.4 µm pore size, SPL Life Sciences, Pocheon, Korea) coated with rat-tail collagen (Corning Inc, Corning, NY, USA). Next, the inserts were suspended in plate wells containing the culture medium after 5 hr of incubation. After 24 hr of incubation, hCMEC/D3 cells were seeded on the apical side of the inserts at $3 \times 10^4$ cells/cm$^2$ and cultured for another 48 hr. Then, the inserts seeded with U251 and hCMEC/D3 cells were suspended in plates seeded with differentiated SH-SY5Y cells and co-cultured for another 6 days. The culture medium was replaced every 24 hr. TEER was periodically measured using a Millicell-ERS (MERS00002) instrument (Millipore, Billerica MA, USA) to monitor cell confluence and development of TJs.

## EdU incorporation assay

The cells were incubated with medium containing 10 µM Edu for 2 hr. Then cells were washed by phosphate-buffered saline (PBS) and harvested by 0.25% trypsin–ethylenediaminetetraacetic acid (#25200072, Gibco, Carlsbad, CA, USA). The EdU incorporation assay was measured using the BeyoClick EdU Cell Proliferation Kit (Beyotime Biotechnology, Shanghai, China) according to the manufacturer's instructions. The samples were determined on the FACSCelesta flow cytometer (Becton, Dickins on and Company, USA), and data were analyzed by Flowjo 10.4 software.

**Table 2.** Initial concentrations in donor chamber and chromatographic conditions of prazosin, verapamil, and lamotrigine.

| Compound | Concentration (µM) | Wavelength (nm) |
|---|---|---|
| Prazosin | 5 | Ex: 250 |
| | | Em: 390 |
| Verapamil | 5 | Ex: 280 |
| | | Em: 310 |
| Lamotrigine | 6 | 220 |

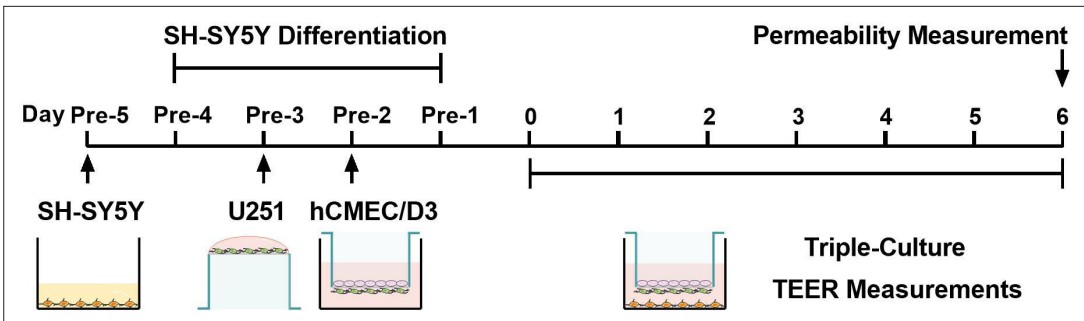

**Figure 9.** Schematic diagram of the establishment process of the triple co-culture blood–brain barrier (BBB) model.

## *In vitro* BBB permeability study

On day 7, the TEER values of BBB models showed a decreasing trend. Therefore, the subsequent experiments were all completed on day 6. The culture medium was removed from the apical and basolateral sides of the inserts and washed twice with preheated Hank's balanced salt solution (HBSS). Fresh HBSS was then added to both the apical and basolateral chambers. After 15 min of preincubation, HBSS in the apical and basolateral chambers was replaced with HBSS containing FITC-dextran 3–5 kDa (FITC-Dex) (Sigma-Aldrich, St. Louis, MO, USA), fluorescein sodium (Sigma-Aldrich, St. Louis, MO, USA), or other tested agents and blank HBSS, respectively. Next, 200 µl aliquots were collected from the basolateral chamber after 30 min of incubation at 37°C. The concentrations of the tested agents in the basolateral chamber were measured.

The apparent permeability coefficient ($P_{app}$, cm/s) values of the tested agents across the *in vitro* BBB model were calculated using the equation (*Tavelin et al., 2002*):

$$P_{app} = (Q/1800)/(S \times C_0) \tag{1}$$

where $S$ is the surface area of the insert membrane (0.33 cm$^2$ for 6.5 mm inserts, 4.46 cm$^2$ for 24 mm inserts), $Q$ is the transported amount of the tested agents transported from the donor chamber to the receiver chamber for 30 min (1800 s), and $C_0$ is the initial concentration of the tested agents in the donor chamber.

## The quantification methods of prazosin, verapamil, lamotrigine

Prazosin, verapamil, and lamotrigine (Aladdin, Shanghai, China) were analyzed by high-performance liquid chromatography (Shimadzu, Kyoto, Japan) with YMC-Triart C18 column (5 µm, 150 × 4.6 mm, YMC America Inc, Allentown, PA, USA). Prazosin and verapamil were detected using the RF-20A fluorescence detector. Lamotrigine was detected using the SPD-20A ultraviolet detector. Samples were centrifugated at 12,000 rpm for 10 min, then 150 µl supernatant was taken and used for analysis. The run temperature was set at 40°C, the injection volume was 20 µl and the flow rate was 1 ml/min. Initial concentrations in donor chamber and other chromatographic conditions of drugs are summarized in *Table 2*.

## The quantification methods of clozapine, venlafaxine, bupropion, amantadine, carbamazepine, fluoxetine, amitriptyline, gabapentin, midazolam, risperidone, olanzapine, mirtazapine, metoclopramide, doxepin, donepezil

All compounds were purchased from Aladdin (Shanghai, China). Except for prazosin, verapamil, and lamotrigine, the other compounds were analyzed by using liquid chromatography–mass spectrometry (Shimadzu, Kyoto, Japan) with YMC-Triart C18 column (5 µm, 150 × 2.0 mm, YMC America Inc, Allentown, PA, USA). Each sample was mixed with 10 µl internal standard. Then 1 ml extraction was added to each sample. The samples were vortex vibrated on the oscillator for 10 min, and then centrifuged at 4°C and 12,000 rpm for 10 min. The supernatant solvent was evaporated with nitrogen flow, then redissolved with 100 µl 40% (vol/vol) acetonitrile, and centrifuged at 4°C and 15,000 rpm for 10 min.

**Table 3.** The summary of mass charge ratio, extraction, initial concentrations in donor chamber.

| Compound | Concentration (µM) | Mass charge ratio [M+H]$^+$ | Extraction |
|---|---|---|---|
| Amantadine | 3 | 181 | Water-saturated N-butanol |
| Amitriptyline | 1.5 | 278 | Ethyl acetate |
| Bupropion | 3 | 240 | Ethyl acetate |
| Carbamazepine | 3 | 237 | Ethyl acetate |
| Clozapine | 4 | 327 | Ethyl acetate |
| Donepezil | 3 | 380 | Methyl tert-butyl ether |
| Doxepin | 4 | 317 | Methyl tert-butyl ether |
| Fluoxetine | 3 | 310 | Ethyl acetate |
| Gabapentin | 10 | 172 | Ethyl acetate |
| Metoclopramide | 4 | 301 | Ethyl acetate |
| Midazolam | 3 | 327 | Ethyl acetate |
| Mirtazapine | 3 | 266 | Methyl tert-butyl ether |
| Olanzapine | 3 | 313 | Methyl tert-butyl ether |
| Risperidone | 4 | 427 | Methyl tert-butyl ether |
| Venlafaxine | 10 | 278 | Ethyl acetate |

The supernatants were injected into LC–MS for analysis. The injection volume of each sample was 5 µl. The mass charge ratio, extraction, and initial concentrations in donor chamber of drugs are summarized in *Table 3*.

## Cell density analysis
On day 6 of co-culture, hCMEC/D3 cells were fixed with 4% paraformaldehyde for 15 min and washed with PBS for three times. Next, the fixed cells were blocked with 5% goat serum for 2 hr and washed with PBS for four times. The blocked cells were incubated with 4',6-diamidino-2-phenylindole (DAPI) (Invitrogen, Carlsbad, CA, USA) and washed with PBS for four times. Cell numbers were counted using Cytation5 (BioTek, Winooski, VT, USA).

## Reverse transcription and qPCR
Total RNA of cells was extracted using RNAiso Plus reagent (Takara Bio Inc, Otsu, Shiga, Japan) and reverse transcribed using HiScript III RT SuperMix (Vazyme, Shanghai, China) as the described method (*Yang et al., 2023*). Paired primers were synthesized by Tsingke Biotech Co., Ltd (Beijing, China), and their sequences are listed in *Table 4*. The SYBR Master Mix was purchased from Yeasen (Shanghai, China). Then, qPCR was performed on the Applied Biosystems QuantStudio 3 real-time PCR system (Thermo Fisher Scientific, Waltham, MA, USA). The mRNA levels of related genes were normalized to *ACTB* or *GAPDH* using the comparative cycle threshold method.

## Western blotting analysis
Whole-cell and tissue lysates, nucleoprotein, and cytoplasmic protein were prepared using RIPA Lysis Buffer (Beyotime, Shanghai, China) as the described method (*Wu et al., 2021*). Proteins were separated through sodium dodecyl sulfate–polyacrylamide gel electrophoresis and transferred onto nitrocellulose or polyvinylidene difluoride membranes. The membranes were blocked with 5% skim milk and incubated with corresponding primary antibodies at 4°C overnight. After being washed with Tris-buffered saline Tween buffer, the membranes were incubated with secondary antibodies (Cell Signaling Technology, MA, USA) at 1:3000 dilution: Anti-mouse IgG, HRP-linked Antibody (#7076), Anti-rabbit IgG, HRP-linked Antibody (#7074). Protein levels were visualized using a highly sensitive

**Table 4.** Primer sequences for quantitative real-time PCR (qPCR) for indicted genes.

| Gene (protein) | Forwards primer, 5′→3′ | Reverse primer, 3′→5′ |
| --- | --- | --- |
| ACTB (β-actin) | GGACTTCGAGCAAGAGATGG | AGCACTGTGTTGGCGTACAG |
| GAPDH (GAPDH) | TGTGGGCATCAATGGATTTGG | ACACCATGTATTCCGGGTCAAT |
| CLDN5 (claudin-5) | CTCTGCTGGTTCGCCAACAT | CAGCTCGTACTTCTGCGACA |
| OCLN (occludin) | ACAAGCGGTTTTATCCAGAGTC | GTCATCCACAGGCGAAGTTAAT |
| TJP1 (ZO-1) | ACCAGTAAGTCGTCCTGATCC | TCGGCCAAATCTTCTCACTCC |
| CDH5 (VE-cadherin) | AAGCGTGAGTCGCAAGAATG | TCTCCAGGTTTTCGCCAGTG |
| ABCB1 (P-gp) | TTGCTGCTTACATTCAGGTTTCA | AGCCTATCTCCTGTCGCATTA |
| ABCG2 (BCRP) | ACGAACGGATTAACAGGGTCA | CTCCAGACACACCACGGAT |
| SLC22A1 (OCT1) | ACGGTGGCGATCATGTACC | CCCATTCTTTTGAGCGATGTGG |
| SLC22A2 (OCT2) | CATCGTCACCGAGTTTAACCTG | AGCCGATACTCATAGAGCCAAT |
| SLC22A8 (OAT3) | ATGGCCCAGTCTATCTTCATGG | GACGGTGCTCAGGGTAATGC |
| SLCO1A1 (OATP1A1) | TAATGTGGGTGTACGTCCTAGT | GCTCCTGTTTCTACAAGCCCAA |
| GDNF (GDNF) | GCAGACCCATCGCCTTTGAT | CCACACCTTTTAGCGGAATGC |
| BDNF (BDNF) | CTACGAGACCAAGTGCAATCC | AATCGCCAGCCAATTCTCTTT |
| NGF (NGF) | TGTGGGTTGGGGATAAGACCA | GCTGTCAACGGGATTTGGGT |
| IGF1 (IGF-1) | GCTCTTCAGTTCGTGTGTGGA | GGTCATGGATGGACCTTACTGT |
| VEGFA (VEGF) | CCCACTGAGGAGTCCAACAT | AAATGCTTTCTCCGCTCTGA |
| FGF2 (bFGF) | AGAAGAGCGACCCTCACATCA | CGGTTAGCACACACTCCTTTG |
| TGFB1 (TGF-β) | GGCCAGATCCTGTCCAAGC | GTGGGTTTCCACCATTAGCAC |

ECL western blotting substrate and a gel imaging system (Tanon Science & Technology, Shanghai, China).

## Preparation of CM

CM of U251 cells (U-CM), SH-SY5Y cells (S-CM), or co-culture of U251 and SH-SY5Y cells (US-CM) was prepared. U251 cells were seeded at the top of the insert membrane and suspended on 6-well plates seeded with differentiated SH-SY5Y cells to co-culture U251 cells with SH-SY5Y cells. The medium was collected every 24 hr. The CMs were subsequently used for hCMEC/D3 cell culture after filtrating with 0.2 µm filters for 144 hr. The levels of GDNF, bFGF, IGF-1, and TGF-β in CMs were measured using corresponding ELISA kits according to the manufacturers' instructions.

## Neutralization of GDNF with anti-GDNF antibody

Exogenous and endogenous GDNF in the medium was neutralized with anti-GDNF antibody (#AF-212-NA, R&D system, Minneapolis, MN, USA). 0.25, 0.5, and 1.0 µg/ml anti-GDNF antibody was added into the US-CM or medium containing 200 pg/ml GDNF, and then the medium was preincubated at 4°C for 1 hr. The hCMEC/D3 cells were incubated with 200 pg/ml GDNF or US-CM containing anti-GDNF antibody or not for 6 days, and then cell lysate was collected for western blot. The medium was replaced every 24 hr.

**Table 5.** The target sequences for small interfering RNA (siRNA) or short hairpin RNA (shRNA).

| Gene | Target sequence |
| --- | --- |
| ETS1 | CGCTATACCTCGGATTACT |
| FOXO1 | AATCTCCTAGGAGAAGAGCTG |
| Gdnf | GCCAGTGTTTATCTGATAC |
| CDH5 | GCCTCTGTCATGTACCAAA |

## Transfection of hCMEC/D3

Here, hCMEC/D3 cells were plated in the plates or culture dishes at $6 \times 10^4$ cells/cm$^2$ and transfected with 10 nM of negative control or human

*FOXO1*, *ETS1,* and *CDH5* siRNA (Tsingke Biotechnology, Beijing, China) for 12 hr using Lipofectamine 3000 (Invitrogen, Carlsbad, CA, USA) reagent according to the manufacturer's instructions. Cells were then incubated with a medium containing GDNF for 72 hr. The siRNA sequences of human *FOXO1*, *ETS1,* and *CDH5* are summarized in *Table 5*.

## FOXO1 overexpression by plasmids

The plasmids encoding FOXO1 (EX-Z7404-M02) were constructed by GeneCopoeia (Rockville, MD, USA). The hCMEC/D3 cells were plated in plates or dishes at $6 \times 10^4$ cells/cm². They were subsequently transfected with 1 μg of negative control or plasmids encoding FOXO1 6 hr using Lipofectamine 3000 reagent according to the manufacturer's instructions. Transfected cells were then incubated with the medium containing GDNF for 72 hr.

## Animals

C57BL/6J mice (male, 4–5 weeks old, 16–18 g, 12 mice) were obtained from Sino-British Sippr/BKLaboratory Animal Ltd (Shanghai, China). Mice were maintained in groups under standard conditions with free access to food and water. Animal studies were performed in accordance with the Guide for the Care and Use of Laboratory Animals (National Institutes of Health) and approved by the Animal Ethics Committee of China Pharmaceutical University (Approval Number: 202307003).

## Brain-specific *Gdnf* knockdown and evaluation of BBB permeability

Mice were randomly divided into control (shNC) and *Gdnf* silencing (sh*Gdnf*) groups (6 mice each group). The shRNA sequence of mice *Gdnf* is listed in *Table 5*. The $2 \times 10^9$ viral genome each of pAAV-U6-shRNA (NC2)-CMV-EGFP or pAAV-U6-shRNA (*Gdnf*)-CMV-EGFP (OBio Technology, Beijing, China) was injected into the bilateral lateral ventricle area (relative to the bregma: anterior–posterior −0.3 mm; medial–lateral ±1.0 mm; dorsal–ventral −3.0 mm) through *i.c.v* infusion. Three weeks following *i.c.v* injection, BBB permeability and expression of corresponding targeted proteins were measured in the mice.

A mixture of FITC-Dex (50 mg/kg) and fluorescein sodium (10 mg/kg) was intravenously administered to experimental mice. Thirty minutes after the injection, the mice were euthanized under isoflurane anesthesia, and brain tissue and plasma samples were obtained quickly. The concentrations of FITC-Dex and fluorescein in the plasma and brain were measured as previously described (*Li et al., 2022b*; *Zhou et al., 2019*). No blinding was performed in animal studies.

## The prediction of drug permeability across BBB using the developed *in vitro* BBB model

The $P_{app}$ values of 18 drugs – prazosin, verapamil, lamotrigine, clozapine, venlafaxine, bupropion, amantadine, carbamazepine, fluoxetine, amitriptyline, gabapentin, midazolam, risperidone, olanzapine, mirtazapine, metoclopramide, doxepin, and donepezil – across the hCMEC/D3 cells mono-culture and triple co-culture models were measured. The predicted *in vivo* permeability-surface area product ($PS_{Pre}$, μl/min/g brain) values across BBB were calculated using the following equation:

$$PS_{Pre} = \frac{P_{app} \times 60 \times VSA \times 1000}{f_{u, brain}}$$

(2)

where *VSA* is the luminal area of the vascular space of brain, which was set to 150 cm²/g (*Fenstermacher et al., 1988*), and $f_{u, brain}$ is the unbound fraction of brain. The published *in vivo* brain permeability values were unified to observed PS ($PS_{Obs}$) by multiplying by *VSA* equal to 150 cm²/g. If $PS_{Pre}$ values were within 0.5- to 2.0-folds of observations, the prediction was considered successful.

## Statistical analyses

All results are presented as mean ± SEM. The average of technical replicates generated a single independent value that contributes to the *n* value used for comparative statistical analysis. The data were assessed for Gaussian distributions using Shapiro–Wilk test. Brown–Forsythe test was employed to evaluate the homogeneity of variance between groups. For comparisons between two groups, statistical significance was determined by unpaired two-tailed *t*-test. The acquired data with significant variation were tested using unpaired *t*-test with Welch's correction, and non-Gaussian

distributed data were tested using Mann–Whitney test. For multiple group comparisons, one-way ANOVA followed by Fisher's LSD test was used to determine statistical significance. The acquired data with significant variation were tested using Welch's ANOVA test, and non-Gaussian distributed data were tested using Kruskal–Wallis test. $p < 0.05$ was considered statistically significant. The simple linear regression analysis was used to examine the presence of a linear relationship between two variables. Data were analyzed using GraphPad Prism software version 8.0.2 (GraphPad Software, La Jolla, CA, USA).

## Acknowledgements

The authors would like to give special thanks to the support of the China Pharmaceutical University Pharmaceutical Animal Laboratory Center.

## Additional information

### Funding

| Funder | Grant reference number | Author |
| --- | --- | --- |
| National Natural Science Foundation of China | 82373943 | Xiaodong Liu |
| National Natural Science Foundation of China | 82173884 | Li Liu |
| National Natural Science Foundation of China | 82204511 | Hanyu Yang |
| Double First Class University Plan | CPU2022QZ21 | Li Liu |
| Jiangsu Funding Program for Excellent Postdoctoral Talent | 2022ZB305 | Hanyu Yang |
| China Postdoctoral Science Foundation | 2023M733897 | Hanyu Yang |
| Postgraduate Research & Practice Innovation Program of Jiangsu Province | KYCX24_1019 | Lu Yang |

The funders had no role in study design, data collection and interpretation, or the decision to submit the work for publication.

### Author contributions

Lu Yang, Conceptualization, Data curation, Funding acquisition, Investigation, Methodology, Writing - original draft, Writing - review and editing; Zijin Lin, Data curation, Formal analysis; Ruijing Mu, Software, Visualization; Wenhan Wu, Formal analysis; Hao Zhi, Data curation; Xiaodong Liu, Conceptualization, Supervision, Funding acquisition, Project administration, Writing - review and editing; Hanyu Yang, Conceptualization, Supervision, Funding acquisition; Li Liu, Conceptualization, Resources, Supervision, Funding acquisition, Project administration

### Author ORCIDs

Lu Yang (ID) https://orcid.org/0000-0003-0032-2045
Li Liu (ID) https://orcid.org/0000-0002-8476-2007

### Ethics

This study was performed in strict accordance with the Guide for the Care and Use of Laboratory Animals (National Institutes of Health). The protocol was approved by the Animal Ethics Committee of China Pharmaceutical University (Approval Number: 202307003). All surgery was performed under isoflurane anesthesia, and every effort was made to minimize suffering.

Reviewer #1 (Public review): https://doi.org/10.7554/eLife.96161.3.sa1
Reviewer #2 (Public review): https://doi.org/10.7554/eLife.96161.3.sa2
Author response https://doi.org/10.7554/eLife.96161.3.sa3

## Additional files

### Supplementary files
- MDAR checklist

### Data availability

All data are available in the manuscript and supporting files; source data files have been provided for Figures 1–6 and Table 1. Source data 1 and Source data 2 files of Figures 1–6 contain original files of western blots. Source data 3 files of Figures 1–6 and Table 1—source data 1 contain the numerical data used to generate the figures.

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
