## [Editor Report · eLife assessment]

This revised study presents **valuable** evidence that a combination of endothelial cells, astrocytes, and neuroblastoma cells of human origin can integrate to form an *in vitro* brain blood barrier, that recapitulates key aspects of its natural counterpart, especially at short times. **Convincingly**, the mechanism by which neuroblastoma-secreted GDNF increases Claudin-5 and VE-cadherin is described. To substantiate the role of GDNF in vivo, authors demonstrated that knock-down of this neurotrophic factor, increased the permeability of the brain blood barrier in mice. This *in vitro* system can be used to study the permeability of the human brain blood barrier to different drugs.

---

## [Referee Report · Reviewer #1 (Public review)]

Summary:

This paper by Yang et al. established an *in vitro* triple co-culture BBB model and demonstrated its advantages compared with the mono or double co-culture BBB model. Further, the authors used their established *in vitro* BBB model and combined it with other methodologies to investigate the specific signaling mechanisms that co-culture with astrocytes but also neurons enhancing the integrity of endothelial cells.

Strengths:

The results persuasively demonstrated that the established triple co-culture BBB model well mimicked several important characteristics of BBB compared with the mono-culture BBB model, including better barrier function and *in vivo*/*in vitro* correlation. The use of human-derived immortalized cells made the model construction process faster and more efficient and had a better *in vivo* correlation without the complications of species differences. This model is expected to be a useful high-throughput evaluation tool for CNS drug development.

Moreover, the authors used a variety of experiments to prove that the triple co-culture model also reflected the interactions between NVU cells, including promoting endothelial cell proliferation and the formation of intercellular junctions. Interestingly, the authors found that neurons also released GDNF to promote barrier properties of brain endothelial cells, as most current research has focused on the promoting effect of astrocytes-derived GDNF on BBB. Meanwhile, the author also validated the functions of GDNF for BBB integrity *in vivo* by silencing GDNF in mouse brains. Overall, the experiments and data presented support the claim that neurons, alongside astrocytes, contribute to the promoting effects of the barrier function of endothelial cells through GDNF secretion.

Weaknesses:

While the authors explained that the use of human-derived immortalized cells has been justified as more reproducible and efficient in constructing the model, the TEER value of the triple co-culture model remains lower than that of the physiological statement. Future research may need to explore additional methods to further enhance the barrier function of the model.

---

## [Referee Report · Reviewer #2 (Public review)]

Summary:

Yang and colleagues developed a new *in vitro* blood-brain barrier model that is relatively simple yet outperforms previous models. By incorporating a neuroblastoma cell line, they demonstrated increased electrical resistance and decreased permeability to small molecules

Strengths:

The authors initially elucidated the soluble mediator responsible for enhancing endothelial functionality, namely GDNF. Subsequently, they elucidated the mechanisms by which GDNF upregulates the expression of VE-cadherin and Claudin-5. They further validated these findings *in vivo*, and demonstrated predictive value for molecular permeability as well. The study is meticulously conducted and easily comprehensible. The conclusions are firmly supported by the data, and the objectives are successfully achieved. This research is poised to advance future investigations in BBB permeability, leakage, dysfunction, disease modeling, and drug delivery, particularly in high-throughput experiments. I anticipate an enthusiastic reception from the community interested in this area. While other studies have produced similar results with tri-cultures (PMID: 25630899), this study notably enhances electrical resistance compared to previous attempts.

Weaknesses:

The power of this system lies in its simplicity, which is enough to study BBB permeability. However, it also lacks some other important cell-cell interactions such as those involving pericytes. Nonetheless, this is still a valuable tool for high throughput screening.

As with many other similar systems, it has lower TEER values compared to the *in vivo* counterpart, this is an issue that researchers in the field will have to address in future studies

---

## [Author Response]

The following is the authors’ response to the original reviews.

**Public Reviews:**

**Reviewer #1 (Public Review):**
Summary:In this manuscript, the authors established an *in vitro* triple co-culture BBB model and demonstrated its advantages compared with the mono or double co-culture BBB model. Further, the authors used their established *in vitro* BBB model and combined it with other methodologies to investigate the specific mechanism that co-culture with astrocytes but also neurons enhanced the integrity of endothelial cells.Strengths:The results persuasively showed the established triple co-culture BBB model well mimicked several important characteristics of BBB compared with the mono-culture BBB model, including better barrier function and *in vivo*/*in vitro* correlation. The human-derived immortalized cells used made the model construction process faster and more efficient, and have a better *in vivo* correlation without species differences. This model is expected to be a useful high-throughput evaluation tool in the development of CNS drugs.Based on the previous experimental results, detailed studies investigated how co-culture with neurons and astrocytes promoted claudin-5 and VE-cadherin in endothelial cells, and the specific signaling mechanisms were also studied. Interestingly, the authors found that neurons also released GDNF to promote barrier properties of brain endothelial cells, as most current research has focused on the promoting effect of astrocytes-derived GDNF on BBB. Meanwhile, the author also validated the functions of GDNF for BBB integrity *in vivo* by silencing GDNF in mouse brains. Overall, the experiments and data presented support their claim that, in addition to astrocytes, neurons also have a promoting effect on the barrier function of endothelial cells through GDNF secretion.Weaknesses:Although the authors demonstrated a highly usable for predicting the BBB permeability, recorded TEER measurements are still far from the human BBB *in vivo* reported measurements of TEER, and expression of transporters was not promoted by co-culture, which may lead to the model being unsuitable for studying drug transport mediated by transporters on BBB.

Thank the reviewer very much for the opportunity to improve our manuscript. The immortalized human cell lines, hCMEC/D3 cell, have poor barrier properties and differences in the expression of some transporters and metabolic enzymes as well as TEER compared to human physiological BBB. However, the use of human primary BMECs may be restricted by the acquisition of materials and ethical approval. Isolation and purification of human primary BMECs are time-consuming and laborious. Moreover, culture conditions can alter transcriptional activity (*PMID*: 37076016). All limit the establishment of BBB models based on primary human BMECs for high-throughput screening. Thus, hCMEC/D3 is still widely used to study characteristics of drug transport across BBB and the effects of certain diseases on BBB (*PMID*: 37076016; 38711118; 31163193) as it is easy to culture and can express a large number of transporters and metabolic enzymes in its physiological state. Therefore, hCMEC/D3 cells were selected to develop our *in vitro* BBB model.

**Reviewer #1 (Recommendations For The Authors):**
Point 1: The authors claim that GDNF is mainly released by human neuroblastoma SH-SY5Y cells in the *in vitro* BBB model, but there are still some differences between the characteristics of cell lines and neurons. The authors should discuss or provide evidence about the distribution and source of GDNF in the brain to support this conclusion.

We greatly appreciate your helpful suggestions. According to your advice, we have revised the “Discussion” in the revised manuscript as follows:

In “Discussion”:

“GDNF is mainly expressed in astrocytes and neurons (Lonka-Nevalaita et al., 2010; Pochon et al., 1997). In adult animals, GDNF is mainly secreted by striatal neurons rather than astrocytes and microglial cells (Hidalgo-Figueroa et al., 2012). The present study also shows that GDNF mRNA levels in SH-SY5Y cells were significantly higher than that in U251 cells. GDNF was also detected in conditioned medium from SH-SY5Y cells. All these results demonstrate that neurons may secrete GDNF”.

Point 2: The authors found that co-culture induced the proliferation of endothelial cells (Figure 1H). I suggest the authors discuss whether the proliferation of endothelial cells would affect their permeability.

Thanks for your suggestion. According to your advice, we have investigated the effect of cell proliferation on the leakage of the cell layer and included the results in Figure 1—figure supplement 1. The present study showed that basic fibroblast growth factor (bFGF) increased cell proliferation of hCMEC/D3 cells but little affected the expression of both claudin-5 and VE-cadherin (in Figure 2F). The hCMEC/D3 cells were incubated with different doses of bFGF and permeabilities of fluorescein (NaF) and FITC-Dextran 3–5 kDa across hCMEC/D3 cell monolayer were measured. The results showed that incubation with bFGF increased cell proliferation and reduced permeabilities of fluorescein and FITC-Dextran across hCMEC/D3 cell monolayer. However, the permeability reduction was less than that by double co-culture with U251 cells or triple co-culture. These results inferred that contribution of cell proliferation to the barrier function of hCMEC/D3 cells was minor. We have made the modifications in “Results” of our manuscript as follows:

In “Result”:

“Furthermore, hCMEC/D3 cells were incubated with basic fibroblast growth factor (bFGF), which promotes cell proliferation without affecting both claudin-5 and VE-cadherin expression (Figure 2F). The results showed that incubation with bFGF increased cell proliferation and reduced permeabilities of fluorescein and FITC-Dex across hCMEC/D3 cell monolayer. However, the permeability reduction was less than that by double co-culture with U251 cells or triple co-culture. These results inferred that contribution of cell proliferation to the barrier function of hCMEC/D3 was minor (Figure 1—figure supplement 1)”.

Point 3: The authors claimed that GDNF induced the expression of claudin-5 and VE-cadherin separately. However, Andrea Taddei et al. reported that VE-cadherin itself also regulates claudin-5 through the inhibitory activity of FoxO1 (Andrea Taddei et al., 2008). The authors did not consider whether the upregulation of claudin-5 is associated with the increase of VE-cadherin.

Thank you for your suggestion. We also investigated whether VE-cadherin affected claudin-5 expression in hCMEC/D3 cells transfected with VE-cadherin siRNA. It was not consistent with the report by Taddei et al. that silencing VE-cadherin only slightly decreased the mRNA level of claudin-5 without significant difference. Furthermore, basal and GDNF-induced claudin-5 protein levels were unaltered by silencing VE-cadherin. The discrepancies may come from characteristics of the tested cells. Endothelial cells derived from murine embryonic stem cells with homozygous null mutation were used in Taddei’s study, while we transfected immortalized brain microvascular endothelial cells with siRNA. Several reports have demonstrated different mechanisms regulating expression of claudin-5 and VE-cadherin. In retinal endothelial cells, hyperglycemia remarkably reduced claudin-5 expression (but not VE-cadherin) (*PMID*: 24594192). However, in hCMEC/D3 cells, hypoglycemia significantly decreased claudin-5 (not VE-cadherin) expression but hyperglycemia increased VE-cadherin expression (not claudin 5) (*PMID*: 24708805). Therefore, the roles of VE-cadherin in regulation of claudin-5 in BBB should be further investigated.

Following your valuable suggestion, we have modified the “Results”, “Discussion” and “Figure 4—figure supplement 1” in the revised manuscript as follows:

In “Result”:

“It was reported that VE-cadherin also upregulates claudin-5 via inhibiting FOXO1 activities (Taddei et al, 2008). Effect of VE-cadherin on claudin-5 was studied in hCMEC/D3 cells silencing VE-cadherin. It was not consistent with the report by Taddei et al. that silencing VE-cadherin only slightly decreased the mRNA level of claudin-5 without significant difference. Furthermore, basal and GDNF-induced claudin-5 protein levels were unaltered by silencing VE-cadherin (Figure 4—figure supplement 1). Thus, the roles of VE-cadherin in regulation of claudin-5 in BBB should be further investigated.”

In “Discussion”:

“Claudin-5 expression is also regulated by VE-cadherin (Taddei et al., 2008). Differing from the previous reports, silencing VE-cadherin with siRNA only slightly affected basal and GDNF-induced claudin-5 expression. The discrepancies may come from different characteristics of the tested cells. Several reports have supported the above deduction. In retinal endothelial cells, hyperglycemia remarkably reduced claudin-5 expression (but not VE-cadherin) (Saker et al., 2014). However, in hCMEC/D3 cells, hypoglycemia significantly decreased claudin-5 expression but hyperglycemia increased VE-cadherin expression (Sajja et al., 2014)”.

“Figure 4—figure supplement 1: The contribution of VE-Cadherin on the GDNF-induced claudin-5 expression. Effects of the VE-Cadherin siRNA (siVE-Cad) on mRNA expression of VE-cadherin (A) and claudin-5 (B). Effects of siVE-Cad and GDNF on claudin-5 and VE-cadherin protein expression (C). NC: negative control plasmids. The above data are shown as the mean ± SEM. Four biological replicates per group. Two technical replicates for A and B, and one technical replicate for C. Statistical significance was determined using unpaired Student’s t-test or one-way ANOVA test followed by Fisher’s LSD test.”

Point 4: The annotation of significance with the p-values in the figures might not be visually concise and clear. It is recommended to provide the p-values in the legends or raw data.

Thank you for your valuable suggestion. We have revised our figures in our revised manuscript. The specific *p*-values and statistical methods were summarized in the source data files of each figure.

Point 5: The authors need to note the material of the Transwell membrane used to increase the reproducibility of experiments, because different materials may cause differences in permeability and TEER (DianeM. Wuest et al., 2013).

We greatly appreciate your valuable suggestions. According to your advice, we have provided the information on the material of the Transwell membrane in the “Materials and Methods” in the revised manuscript as follows:

In “Materials and Methods”:

“U251 cells were seeded at 2 × 10^4^ cells/cm^2^ on the bottom of Transwell inserts (PET, 0.4 µm pore size, SPL Life Sciences, Pocheon, Korea) coated with rat-tail collagen (Corning Inc, Corning, NY, USA)”.

Point 6: It is not necessary to abbreviate "*in vitro*/*in vivo* correlation" in the legend of Figure 7 as it was not mentioned again in the following text.

Thank you for your valuable suggestion. We have deleted the abbreviation of "Figure 7" of the revised manuscript.

In “Figure 7”

“Figure 7. *In vitro*/*in vivo* correlation assay of BBB permeability."

**Reviewer #2 (Public Review):**
Summary:Yang and colleagues developed a new *in vitro* blood-brain barrier model that is relatively simple yet outperforms previous models. By incorporating a neuroblastoma cell line, they demonstrated increased electrical resistance and decreased permeability to small molecules.Strengths:The authors initially elucidated the soluble mediator responsible for enhancing endothelial functionality, namely GDNF. Subsequently, they elucidated the mechanisms by which GDNF upregulates the expression of VE-cadherin and Claudin-5. They further validated these findings *in vivo*, and demonstrated predictive value for molecular permeability as well. The study is meticulously conducted and easily comprehensible. The conclusions are firmly supported by the data, and the objectives are successfully achieved. This research is poised to advance future investigations in BBB permeability, leakage, dysfunction, disease modeling, and drug delivery, particularly in high-throughput experiments. I anticipate an enthusiastic reception from the community interested in this area. While other studies have produced similar results with tri-cultures (PMID: 25630899), this study notably enhances electrical resistance compared to previous attempts.Weaknesses:(A) Considerable effort has been directed towards developing *in vitro* models that more closely resemble their *in vivo* counterparts, utilizing stem cell-derived NVU cells. Although these examples are currently rudimentary, they offer better BBB mimicry than Yang's study.

Thank you very much for your valuable comments. Indeed, hCMEC/D3 cells, have poor barrier properties and low TEER compared to human physiological BBB. The human pluripotent stem cells BBB models (hPSC-BBB models) make it possible to provide a robust and scalable cell source for BBB modeling, although many challenges remain, particularly concerning reproducibility and recreation of multifaceted phenotypes *in vitro* with increasing complexity. Moreover, the hPSC-derived BBB models are highly dependent upon the heterogeneous incorporation of hPSC-derived BMEC origins, cells derived from different protocols are not well validated and standardized in the BBB models. Thus, the hPSC-BBB models are still being developed and their clinic applications are still at an early stage (*PMID*: 34815809; 35755780). The hCMEC/D3 cell line is still widely used to study characteristics of drug transport across BBB and the effects of certain diseases on BBB (*PMID*: 37076016; 38711118; 31163193) as it is easy to culture and can express a large number of transporters and metabolic enzymes in its physiological state. Therefore, hCMEC/D3 cells were selected to develop our *in vitro* BBB model.

(B) Additionally, some instances might benefit from more robust statistical tests; nonetheless, I do not think this would significantly alter the experimental conclusions.

Thank you for your valuable suggestions on the statistical methods used in our study, which made us realize our lack of rigor in selecting statistical methods. We have made modifications to statistical methods, and all statistical results showed the manuscript have been updated accordingly.

(C) Similar experiments with tri-cultures yielding analogous results have been reported by other authors (PMID: 25630899). TEER values are a bit higher than the aforementioned experiments; however, this study has values at least one order of magnitude lower than physiological levels.

Thank your advice. We also noticed that TEER values in the present study were different from previous reports, which may come from types of BEMCs, astrocytes, and neurons.

**Reviewer #2 (Recommendations For The Authors):**
Point 1: If you've already decided to enhance the model by incorporating additional cell types, why not include pericytes as well? As mentioned in the public review, other studies have explored tri-culture models; adding pericytes or other cell types could provide valuable insights.

We greatly appreciate your helpful suggestions. As you mentioned, the barrier function of our model still needs further improvement, which is also a limitation of our current model. In our future research, we will aim to optimize our model by incorporating other NVU cells. Beyond drug screening, we also hope that our *in vitro* BBB model can serve as a versatile tool to investigate underlying factors associated with neuropathological disorders. According to your advice, we have modified “Discussion” in the revised manuscript as follows:

In “Discussion”:

“However, the study also has some limitations. In addition to neurons and astrocytes, other cells such as microglia, pericytes, and vascular smooth muscle cells, especially pericytes, may also affect BBB function. How pericytes affect BBB function and interaction among neurons, astrocytes, and pericytes needs further investigation.”

Point 2: The decline in TEER after 6 days is concerning. Have you extended your experiments beyond day 7? If so, what were the outcomes? Did the system degrade, leading to decreased resistance, or did cell death occur?

We greatly appreciate your helpful recommendation. We also observed that the TEER of our culture system began to decline on day 7. To ensure the reliability of our experiments, our experiments were conducted on day 6 of co-cultivation and did not extend beyond day 7. We speculate that the reason for the decrease in TEER values may be due to excessive cell contact, which could inhibit cell proliferation and long-term cultivation may lead to cell aging. Similar results showing a decrease in TEER of *in vitro* BBB models after prolonged culture have been reported in other studies (*PMID*: 31079318; 8470770). To eliminate misunderstandings, we have made the following modifications to our manuscript:

In “Result”:

“TEER values were measured during the co-culture (Figure 1B)*.* TEER values of the four *in vitro* BBB models gradually increased until day 6. On day 7, the TEER values showed a decreasing trend. Thus, six-day co-culture period was used for subsequent experiments”.

In “*In vitro* BBB permeability study” of “Materials and Methods”:

“On day 7, the TEER values of BBB models showed a decreasing trend. Therefore, the subsequent experiments were all completed on day 6”.

Point 3: It is standard practice for figures to be referenced in the order they appear in the manuscript. However, Figures 1A and 1B are not mentioned until the end of the methods section. Adding a brief sentence at the beginning of the main body referencing these figures would improve the clarity of the experimental approach.

Thank you for your valuable suggestion. We had made modifications to Figure 1, and the details of the cell model establishment process had been included in Figure 9 which is mentioned in the “Materials and Methods” section.

Point 4: To strengthen the evidence supporting the proliferative effect of GDNF, consider incorporating additional measures beyond cell count alone. While an increase in cell count could be attributed to reduced cell death (given GDNF's pro-survival properties), proliferation effects have also been shown (PMID: 28878618). I suggest demonstrating proliferation with markers or cell cycle analysis would provide more robust evidence.

Thank you for your helpful suggestion. We used EdU incorporation and CCK-8 assays to further detect the proliferation of hCMEC/D3 cells, and corresponding results were added in the revised Figure 1H and Figure 1I. The description of results is shown as follows:

In “Results”:

“Co-culture with SH-SY5Y, U251, and U251 + SH-SY5Y cells also enhanced the proliferation of hCMEC/D3 cells. Moreover, the promoting effect of SH-SY5Y cells was stronger than that of U251 cells (Figure 1G-I).”

Point 5: Could you specify the use of technical replicates in your experiments? How many?

Thank you for your helpful suggestion, and we apologize for the issue you pointed out. We have now specified the technical replicates of experiments in the legends of the revised manuscript. In general, the technical replicate number of ELISA and qPCR is two, and that of the rest experiments is one. And we have also made the following modifications to our manuscript:

In “Statistical analyses” of “Materials and Methods”:

“All results are presented as mean ± SEM. The average of technical replicates generated a single independent value that contributes to the n value used for comparative statistical analysis”.

Point 6: Given the sample size of 4 in most experiments, it may be insufficient for passing a normality test. Therefore, it's advisable to employ non-parametric tests such as the Kruskal-Wallis test, followed by appropriate post-hoc tests.

Thank you for your valuable and useful suggestion. We apologize for our initial oversight regarding statistics. Based on your suggestion, we have thoroughly reviewed and revised the statistical methods and statistical results in the manuscript. Referring to the ‘Statistics Guide’ of GraphPad (H. J. Motulsky, "The power of nonparametric tests", GraphPad Statistics Guide. Accessed 20 June 2024. https://www.graphpad.com/guides/prism/latest/statistics/stat_the_power_of_nonparametric_tes.htm), the Kruskal-Wallis test is more robust when the data does not follow a normal distribution or homogeneity of variance. However, due to its reliance on ranks, it may have lower sensitivity in detecting small differences. If the total sample size is tiny, the Kruskal-Wallis test will always give a P value greater than 0.05 no matter how much the groups differ. To address this, we first used the Shapiro-Wilk test to assume whether the samples come from Gaussian distributions. For samples meeting this criterion, parametric tests were employed. For samples that do not follow the Gaussian distribution, as per your advice, we utilized the non-parametric tests. We have modified the “Statistical analyses” in the revised manuscript as follows:

In “Statistical analyses” of “Materials and Methods”:

“The data were assessed for Gaussian distributions using Shapiro-Wilk test. Brown-Forsythe test was employed to evaluate the homogeneity of variance between groups. For comparisons between two groups, statistical significance was determined by unpaired 2-tailed t-test. The acquired data with significant variation were tested using unpaired t-test with Welch's correction, and non-Gaussian distributed data were tested using Mann-Whitney test. For multiple group comparisons, one-way ANOVA followed by Fisher’s LSD test was used to determine statistical significance. The acquired data with significant variation were tested using Welch's ANOVA test, and non-Gaussian distributed data were tested using Kruskal-Wallis test. p < 0.05 was considered statistically significant. The simple linear regression analysis was used to examine the presence of a linear relationship between two variables. Data were analyzed using GraphPad Prism software version 8.0.2 (GraphPad Software, La Jolla, CA, USA)”.